# Practical Bayesian Optimization of Objectives with Conditioning Variables

## Abstract

Bayesian optimization is a class of data efficient model based algorithms typically focused on global optimization. We consider the more general case where a user is faced with multiple tasks that each need to be optimized, find the global optima within each task, all the task conditional optima. For example given a range of cities with different patient distributions, we optimize the ambulance locations for each and every city; given subclass partitions of CIFAR-10, we optimize CNN hyperparameters for each partition. Similarity across tasks boosts optimization of each task in two ways: in modelling by data sharing across objectives, and also in acquisition by quantifying how a single point on one task can help learn the optima of similar tasks. For this we propose a framework for conditional optimization: ConBO. This can be built on top of a range of acquisition functions and we propose a new Hybrid Knowledge Gradient acquisition function. The resulting method is intuitive and theoretically grounded, either matches or significantly outperforms recently published works on a range of problems, and thanks to the unique nature of conditional optimization, is easily parallelized to collect a batch of points.

## 1 Introduction

Expensive stochastic black box functions arise in many fields such as fluid simulations [1], engineering wing design [2], and machine learning parameter tuning [3]. Bayesian optimization is a powerful set of tools to optimize such functions, finding the input with highest long term average performance $x^* = \arg\max_x \mathbb{E}[f(x)]$, (the expectation represents averaging over the performance noise). In this work we consider an under-explored generalization of the standard setting previously referred to as "conditional optimization" [4] where a user has a collection of functions, or *tasks*, and simply seeks the peak of each function/task. Formally, we have an expensive black box $f$ that takes as arguments both a *task* $s \in S$ and an *input* $x \in X$, typically box-constrained continuous variables, and returns a noisy scalar performance

$$f(s, x) : S \times X \to \mathbb{R}. \tag{1}$$

At each iteration an algorithm determines both task and input $(s, x)$ then observes performance $y = f(s, x)$ and the goal is to learn the input with highest average performance for each task

$$x^*(s) = \arg\max_x \mathbb{E}[f(s, x)]. \tag{2}$$

$x^*(s)$ is referred to as the optima conditioned on $s$, see Figure 1 black line. In certain applications, one may want to give higher priority to particular tasks hence a task weighting function $W(s)$ may also be specified. In this work we consider the following applications.

**CNN hyperparameters:** the CIFAR-10 dataset contains 10 classes, this is split into five mutually exclusive binary classification datasets and for each we train a CNN, each CNN is a task in $S =$

$\{1, ..., 5\}$. For each CNN, we optimize dropout rates, batch size and Adam parameters, so $X \subset \mathbb{R}^7$ and $f(s, x)$ is the validation accuracy. We assume all five CNNs have equal priority, $W(s) = 1/5$.

**Ambulances in a square:** [5] given a range of 30km×30km cities, each city is a task with a different population centre $s \in [0, 30]^2$. For a given city, we optimize the 2D location of three ambulance bases, $x \in [0, 30]^{3 \times 2} \subset \mathbb{R}^6$. Given a city $s$ and ambulance bases $x$, a virtual environment randomly generates patients for a simulated day and the average ambulance journey time, $f(s, x)$, is returned. Inland cities, like Paris or London, with population centres in the middle are more common than coastal cities, like Singapore or Dubai, with a population centre on a boundary. Thus, $W(s)$ mirrors this distribution of city centres, a Gaussian centred at $s = (15, 15)$ with 7km standard deviation and truncated to $[0, 30]^2$.

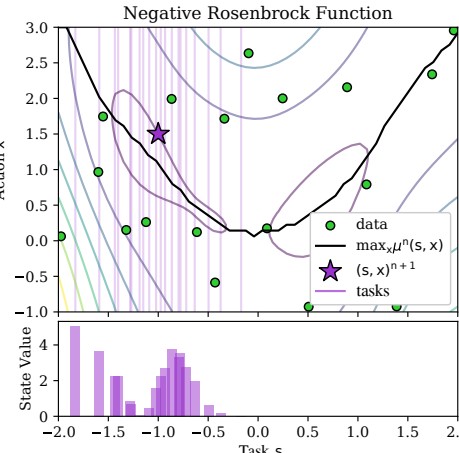

Figure 1: Top: GP model. Each task (vertical slice) is an objective function to maximise. A new sample $(s, x)^{n+1}$ provides information about the optima of similar tasks. Bottom: the acquisition function value of each task using hybrid Knowledge Gradient.

**Assemble to order:** [6, 7] a company owns many stock warehouses and each one faces a different level of demand $s \in [0.5, 1.5]$. Equal priority is given to all warehouses and so $W(s)$ is uniform. At each warehouse, stock levels are controlled by setting targets of a control policy $x = [0, 20]^8$. Given a demand level $s$ and control parameters $x$, a simulator generates customer orders according to demand, the warehouse stock is sold and replenished according to policy $x$, and profit over a simulated month is returned as $f(s, x)$.

The closely related multi-task Bayesian optimisation or Bayesian Quadrature optimisation methods [8, 9, 10, 11] aim to find a single peak that maximises the weighted sum/integral over tasks, $x^* = \arg\max_x \int \mathbb{E}[f(s, x)]W(s)ds$. However, as they are fundamentally designed for a different purpose, we observe that they struggle to find the peak of each and every task, see experiments in Supplementary Material 6 (SM 6).

Also closely related are contextual optimization algorithms [12, 13, 14], conditional optimization has also been referred to as "offline contextual" [15]. In contextual applications, at iteration $n$ the next task (called context) $s^n$ is passed from the black box to a contextual algorithm that intelligently determines $x^n$. Then the black box returns both performance $y^n$ and the next task/context $s^{n+1}$. One could "hotfix" a contextual algorithm for conditional optimization problems by randomly choosing $s^{n+1} \sim \mathbb{P}[s] \propto W(s)$ at each iteration. However, we empirically show that, unsurprisingly, randomly choosing the task in each iteration is significantly worse than intelligently choosing the task.

Multi-fidelity and multi-information source methods [16, 17, 18, 19] assume there is a single known constant target task $s^* \in S$ corresponding to the highest fidelity level or most expensive information source that must be optimized, $x^* = \arg\max_x \mathbb{E}[f(s^*, x)]$. If the cost of evaluation varies across the function domain, $c(s, x)$, cheaper regions of the domain can be evaluated improving sample efficiency. Again, such methods may be hotfixed for the setting we consider by artificially designating a target task, e.g. the highest priority task $s^* = \arg\max_s W(s)$. However, we observe that if cost is constant across the domain (as is commonly assumed in the BO literature), these methods greedily optimise the artificial target task $s^*$ and blindly neglect all other tasks, see SM4. This desirable behaviour in multi-fidelity optimization problems leads to failure in conditional optimization problems.

To the best of our knowledge, there exist only a small number of works that propose algorithms specifically designed for the conditional setting.

The Surrogate Collaborative Tuning (SCoT) algorithm [20] optimizes a finite set of tasks. It iteratively visits each task, $s$, in a round-robin fashion and determines the input $x$ by expected improvement (EI). The authors apply this to optimize the hyperparameters of an ML model for multiple datasets. The

Profile Expected Improvement (PEI) and Profile Expected Quantile Improvment (PEQI) algorithms [21, 4] consider continuous tasks and significantly improve upon SCoT by dynamically determining the task in each iteration. The acquisition value of a given point $(s, x)$ is the expected improvement of a new output over the best predicted output within the same task.

The REVI algorithm [22] at each iteration discretizes both task $S$ and input $X$ spaces. Given a new hypothetical point, $(s, x)^{n+1}$, for each task in the discretization, $s_i$, the discrete Knowledge Gradient over the $X$ discretization quantifies how $(s, x)^{n+1}$ will benefit task $s_i$. Summing the benefits over the tasks yields the acquisition value of $(s, x)^{n+1}$. REVI was designed to account for how all tasks can be optimized by each sample. However, this comes with exponential cost, increasing problem dimension leads to exponentially increasing discretization size with corresponding exponential space and time requirements. On the other hand, sparse discretizations can lead to poor and arbitrary measures of acquisition value. Consequently, REVI has only successfully been applied to low-dimensional synthetic problems.

Most recently, the Multi-task Thompson sampling (MTS) method [15] uses a novel kernel with a length scale that varies across tasks. This is combined with a Thompson sampling method for collecting new data and showed performance improvements over REVI. However, Thompson sampling also does not account for how one sample provides benefit for optimizing all tasks, a fundamental structural property of conditional optimization.

We propose a method that exploits the basic structure of conditional optimization while also being highly scalable, and therefore applicable to more challenging real-world problems. We make the following contributions:

1. A framework for conditional Bayesian optimization with theoretical guarantees: ConBO.
2. A new, fast global optimization method: Hybrid Knowledge Gradient.
3. State-of-the-art performance on open source problems including CNNs and simulators.

## 2   The Conditional Bayesian Optimization Algorithm

We first discuss the fitting of the Gaussian process model and the predicted conditional optima. We then motivate the acquisition value for a single task and how this is integrated over tasks yielding the acquisition function. Because integrating over tasks multiplies the computational burden, we propose Hybrid Knowledge Gradient as a solution. See Algorithm 1 in the SM 2 for a high level summary.

At a stage after having observed $n$ data points, $\{(s^i, x^i, y^i)\}_{i=1}^n$ where $y^i = f(s^i, x^i)$, a Gaussian process is fit over the joint space $S \times X$ to scalar outputs $y$. Let $\tilde{X}^n = ((s, x)^1, ..., (s, x)^n)$ and $Y^n = (y^1, ..., y^n)$. A Gaussian process is defined by a prior mean and prior covariance function, $\mu^0(s, x)$, $k^0((s, x), (s', x'))$ which are chosen for each application, for more information see [23]. Let the data covariance matrix be $K = k^0(\tilde{X}^n, \tilde{X}^n) \in \mathbb{R}^{n \times n}$, the posterior mean is given by

$$\mu^n(s, x) = \mu^0(s, x) + k^0((s, x), \tilde{X}^n)(K + \sigma_0^2 I)^{-1}(Y^n - \mu^0(\tilde{X}^n))$$

and the posterior covariance is given by

$$k^n((s, x), (s', x')) = k^0((s, x), (s', x')) + k^0((s, x), \tilde{X}^n)(K + \sigma_0^2 I)^{-1} k^0(\tilde{X}^n, (s', x')).$$

At the end of data collection, in standard BO, often the peak posterior mean is returned as the predicted best input. Generalizing to the conditional setting we simply condition on $s$,

$$x^{*N}(s) = \arg\max_x \mu^N(s, x). \tag{3}$$

During data collection, a new data point $y^{n+1}$ at $(s, x)^{n+1}$ will update the Gaussian process over the whole domain $S \times X$. To construct an acquisition function for conditional optimization, we start by looking for standard acquisition functions that account for how the model changes at *unsampled* locations $(s', x') \neq (s, x)^{n+1}$. Specifically, the popular Expected improvement (EI) [24] and upper confidence bound (UCB) [25] methods are both functions of the mean and kernel *at the sampled point only*, i.e. of $\left(\mu^n((s, x)^{n+1}), k^n((s, x)^{n+1}, (s, x)^{n+1})\right)$, hence would require non-trivial modification to be able to account for how a sample affects similar tasks. Methods that *do* utilise the mean and kernel at unsampled points include Entropy search (ES [26, 27] and PES [28]) that measures the

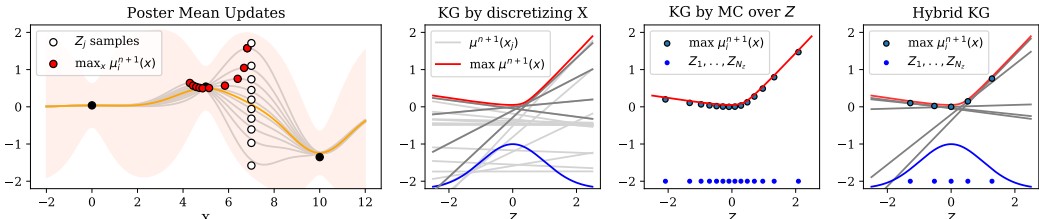

Figure 2: Methods for computing $KG(x^{n+1})$ at $x^{n+1} = 7$. Left: $\mu^n(x)$ and samples of $\mu^{n+1}(x)$ determined by a scalar $Z \sim N(0,1)$. Centre-left: $KG_d$ replaces $X$ with up to 3000 points $x_i \in X_d$ and $\mu^{n+1}(x_i)$ is linear in $Z$. Centre-right: $KG_{MC}$ samples up to 1000 functions $\mu^{n+1}(x)$ functions and maximises each of them numerically. Right: $KG_h$ samples up to 5 functions $\mu^{n+1}(x)$ and maximizes them numerically, the $\arg\max$ points $x_1^*, .., x_5^*$ are used as $X_d$ in $KG_d$.

mutual information between the new output $\mathbb{P}[y^{n+1}|x^{n+1}]$ and the (uncertain) location of the peak $\mathbb{P}[x^*|\tilde{X}^n, Y^n]$, Max-value entropy search (MES) [29] that measures mutual information between the new output $\mathbb{P}[y^{n+1}|x^{n+1}]$ and the (uncertain) largest output $\mathbb{P}[\max y|\tilde{X}^n, Y^n]$, and Knowledge Gradient (KG) [30] that measures the expected peak of the new posterior mean $\mathbb{E}[\max \mu^{n+1}(x)]$ caused by a new $y^{n+1}$ at $x^{n+1}$.

Each single task $s$ defines a single global optimization problem over $x \in X$. Given a proposed sample $(s,x)^{n+1} = (s^{n+1}, x^{n+1})$, the acquisition benefit for task $s$ may be computed using ES, PES, MES or KG. In this work we adopt KG for its Bayesian decision theoretic derivation that extends seamlessly to the conditional setting. For KG, the benefit for a given task is the expected increase in peak predicted performance within the task. We denote the task-conditioned KG as

$$KG_c(s; (s,x)^{n+1}) = \mathbb{E}_{y^{n+1}}[\max_{x'} \mu^{n+1}(s, x')|(s,x)^{n+1}] - \max_{x''} \mu^n(s, x'').$$

We discuss numerical evaluation of $KG_c(\cdot)$ in Section 2.1. Similar expressions for conditioned entropy methods, $ES_c(\cdot)$, $PES_c(\cdot)$, $MES_c(\cdot)$, are derived in SM 7. Integrating over all tasks $s$ yields the total acquisition value

$$\int_S KG_c(s; (s,x)^{n+1})W(s)ds. \tag{4}$$

For discrete $S$ the integral is replaced by summation. For continuous $S$, the integral over tasks $s$ cannot be computed analytically so we use Monte Carlo with importance sampling. When using a kernel that factorises $k(s,x,s',x') = \sigma_0^2 k_S(s,s')k_X(x,x')$, like squared exponential or Matérn, similarity across tasks is encoded in $k_S(s,s')$. This naturally leads to the proposal distribution $q(s|s^{n+1}) \propto k_S(s,s^{n+1})$. In our continuous task experiments, we use the Matérn kernel and a Gaussian proposal distribution with mean $s^{n+1}$ and the task kernel length scales, $l_s$, as standard deviations,

$$q(s|s^{n+1}) \sim \mathcal{N}(s|s^{n+1}, \text{diag}(l_s^2)). \tag{5}$$

We generate $n_s = 20$ samples $S_{MC} = \{s_1, ..., s_{n_s}\}$, finally the acquisition function is

$$\text{ConBO}(s^{n+1}, x^{n+1}) = \sum_{s_i \in S_{MC}} \frac{W(s_i)}{q(s_i|s^{n+1})} KG_c(s_i; (s,x)^{n+1}).$$

Figure 1 shows a set of sampled tasks and the $KG_c(\cdot)$ for each one. Each KG term directly measures increase in predicted performance for one task, e.g. if $y$ values are dollar amounts, ConBO with $KG_c(\cdot)$ is the sum of dollar increases over all tasks. However, for entropy based methods, ConBO becomes a sum of Shannon information units thereby indirectly optimizing the dollar amounts. The randomly sampled tasks $S_{MC}$ may be resampled with each call to $\text{ConBO}(s,x)$ and gradients estimated enabling the optimal $(s,x)^{n+1}$ to be found with a stochastic gradient ascent optimizer such as Adam [31]. Selecting each point according to maximising ConBO is also myopically optimal in a value of information framework:

**Theorem 1** *Let $(s^*, x^*) \in \arg\max ConBO(s,x)$ be a point chosen for sampling. $(s^*, x^*)$ is also the point that maximises the myopic Value of Information, the increase in predicted performance.*

Further, in finite search space, with an infinite sampling budget all points will be sampled infinitely:

**Algorithm 1** Computing ConBO$(s, x)$. For each call with a candidate (task, input) point, similar tasks are sampled. For each sampled task a cheap acquisition function, hybrid KG, is evaluated. The output is the importance weighted average of hybrid KG values. Details and full numerical expressions are given in SM 3.

---

**Require:** candidate point $(s, x)^{n+1}$, parameters $n_s$, $n_z$
    Sample $n_s$ tasks similar to $s^{n+1}$, $S_{MC} \sim q(s|s^{n+1})$
    Initialize output value $Q \leftarrow 0$
    **parfor** $s_i$ **in** $S_{MC}$ **do**

>     Initialize $\tilde{X}_i^* \leftarrow \{\}$          *Hybrid Knowledge Gradient*
>     **parfor** $j$ **in** $1, ..., n_z$ **do**
>         $z_j \leftarrow \Phi^{-1}\big((2j - 1)/2n_z\big)$
>         $y_j^{n+1}$ computed with $(s, x)^{n+1}$ and $z_j$
>         $\mu_j^{n+1}(s, x)$ constructed by rank-1 update with $(s, x, y_j)^{n+1}$
>         $x_j^* \leftarrow \arg\max_x \mu_j^{n+1}(s_i, x)$ with `Optimizer()`
>         $\tilde{X}_i^* \leftarrow \tilde{X}_i^* \cup (s_i, x_j^*)$
>     **end parfor**
>     $\alpha_i \leftarrow \mathrm{KG}_d(\tilde{X}_i^*)$

        $Q \leftarrow Q + \alpha_i\, W(s_i)\,/\,q(s_i|s^{n+1})$
    **end parfor**
    **return** average over tasks $Q/n_s$

---

**Theorem 2** *Let $S$ and $X$ be finite sets and $N$ the budget to be sequentially allocated by ConBO. Let $n(s, x, N)$ be the number of samples allocated to a point $(s, x)$ within budget $N$. Then for all $(s, x) \in S \times X$ we have that $\lim_{N \to \infty} n(s, x, N) = \infty$.*

The law of large numbers ensures that the algorithm learns the true expected performance for all points. Proofs are given in the SM 1.

## 2.1 Hybrid Knowledge Gradient

By definition, KG is more expensive than EI and UCB. Further, the function $\mathrm{KG}_c(s_i, (s, x)^{n+1})$ must be computed once for each sampled task $s_i$, the computational cost is therefore $n_s$ times the global acquisition function equivalent. To alleviate this cost, we propose a novel, efficient algorithm for computing KG. In the following section we assume constant $s$ for brevity, reducing to the global optimization setting. Given a hypothetical location $x^{n+1}$, KG quantifies the value of a new hypothetical observation $y^{n+1}$ by the expected increase in the peak of the posterior mean

$$\mathrm{KG}(x^{n+1}) = \mathbb{E}_{y^{n+1}}\big[\max_{x'} \mu^{n+1}(x')\big|x^{n+1}\big] - \max_{x''} \mu^n(x''). \tag{6}$$

However, $\max_{x'} \mu^{n+1}(x')$ has no explicit formula and approximations are required which we describe next. At time $n$, the next posterior mean is unknown, however, it may be written as $\mu^{n+1}(x) = \mu^n(x) + \tilde{\sigma}(x; x^{n+1})Z$ where $\tilde{\sigma}(x; x^{n+1})$ is a deterministic function and the scalar $Z \sim \mathcal{N}(0, 1)$ captures the randomness of $y^{n+1}$, see SM 2. Previously, $\mathrm{KG}(x)$ has been computed in two ways.

**KG by discretization** [30, 32]: in Equation 6, the maximizations may be performed over a discrete set of $d$ points $X_d \subset X$. Denoting $\underline{\mu} = \mu^n(X_d) \in \mathbb{R}^d$ and $\underline{\tilde{\sigma}}(x^{n+1}) = \tilde{\sigma}(X_d; x^{n+1}) \in \mathbb{R}^d$, then

$$\mathrm{KG}_d(x^{n+1}) = \mathbb{E}_Z\big[\max\{\underline{\mu} + \underline{\tilde{\sigma}}(x^{n+1})Z\}\big] - \max \underline{\mu}.$$

The $\max\{\underline{\mu} + \underline{\tilde{\sigma}}(x^{n+1})Z\}$ is a piece-wise linear function of $Z$ and the expectation is analytically tractable. The output is a *lower bound* of the true $\mathrm{KG}(x)$. REVI [22] and the MiSo algorithm [17] used $\mathrm{KG}_d$ with 3000 uniformly random distributed points. This method suffers the curse of dimensionality as $X_d$ must grow exponentially with dimension. Further, the discretization may likely contain many useless points in uneventful regions of $X$, see Figure 2 centre-left plot.

| KG | Type of Estimate | $n_z = 3$ | $n_z = 5$ | $n_z = 7$ | $n_z = 50$ |
|---|---|---|---|---|---|
| Discrete | lower bound | 0.24 (1.37) | 0.41 (1.49) | 0.55 (1.91) | 2.03 (2.2) |
| MC | unbiased | 3.54 (4.79) | 3.73 (4.51) | 3.50 (3.39) | 3.36 (1.18) |
| MC++ | unbiased | 2.97 (3.78) | 3.50 (2.68) | 3.22 (1.69) | 3.32 (0.2) |
| Hybrid | lower bound | 3.15 (0.00) | 3.28 (0.00) | 3.31 (0.00) | 3.34 (0.00) |

Table 1: We collect 20 points on the Rosenbrock function, a point was randomly selected and KG computed by the different methods. Each calculation is repeated 50 times and we report mean and two standard deviations. The Monte Carlo methods suffer from high variance, the discrete method is volatile and returns a very loose lower bound. Hybrid KG is very stable with errors too small to show.

**KG by Monte Carlo** [33, 34]: given $x^{n+1}$, the method samples up to $n_z = 1000$ Gaussian values of $Z$. For each $Z_j$, construct the posterior mean, $\mu_j^{n+1}(x)$, find the maximum with a continuous numerical `Optimizer()` like L-BFGS or CG. Averaging the maxima from all $Z_j$ yields

$$\text{KG}_{MC}(x^{n+1}) = \frac{1}{n_s} \sum_j \underset{x'}{\text{Optimizer}}\big(\mu_j^{n+1}(x)\big) - \underset{x''}{\text{Optimizer}}\big(\mu^n(x'')\big).$$

The result is an *unbiased* estimate of true $KG(x)$ and scales better to higher dimensional $X$, the univariate $Z$ is discretized by Monte Carlo samples instead of $X$. However, for a good estimate, $n_z$ must be large, many `Optimizer()` calls are required. See Figure 2 centre right.

We instead propose a simple mixture of the two approaches above that both scales to higher dimensional $X$ and drastically reduces the number of `Optimizer()` calls.

**Hybrid KG:** given $x^{n+1}$, following $\text{KG}_{MC}$ we use $n_z = 5$ values of $Z_j$, construct $\mu_j^{n+1}(x)$ and use `Optimizer()` to find the peak location $x_j^*$. The set of peak locations $X_d^* = \{x_1^*, ..., x_{n_z}^*\}$ is used as a *dynamic optimized* discretization in $\text{KG}_d$ thus analytically computing an extremely tight lower bound of the true $KG(x)$. Let $\underline{\mu}^* = \mu^n(X_d^*)$ and similarly for $\underline{\tilde{\sigma}}^*(x^{n+1})$, Hybrid KG is given by

$$\text{KG}_h(x^{n+1}) = \mathbb{E}_Z[\max \underline{\mu}^* + \underline{\tilde{\sigma}}^*(x^{n+1})Z] - \max \underline{\mu}^*.$$

Compared to $\text{KG}_d$, the hybrid method removes redundant points in the discretization $X_d$, all $X_d^*$ points contribute to $\max \mu^{n+1}(X_d^*)$ and there are far fewer points. Compared to $\text{KG}_{MC}$ that samples many $Z_j$ and optimizes many $x_j^*$, Hybrid KG uses far fewer $Z_j$ and optimizes far fewer $x_j^*$. See Algorithm 1 for a summary of evaluating ConBO with Hybrid KG. ConBO can be reduced to the REVI algorithm by replacing the dynamic importance sampled tasks with a pre-frozen discretization of tasks and the (dynamic optimized) hybrid KG with discrete KG over a pre-frozen discretization of inputs. These changes drastically improve scalability enabling ConBO to be applied to a far broader range of applications.

To ensure asymptotic convergence, in a discrete domain, we require that the acquisition function is non-negative, $\text{KG}_h(x) \geq 0$, and the acquisition function is zero where GP variance is zero, $\text{KG}_h(x) = 0 \iff k^n(x, x) = 0$. Therefore, always choosing $x^{n+1} = \arg\max \text{KG}_h(x)$ ensures only points with GP variance will be revisited until all points have no variance i.e. the true function is known for all points. We can ensure these properties by setting $n_z \geq 2$ and at least one $Z_j$ is equal to zero.

**Theorem 3** *Let $n_z \geq 2$ and let $\underline{Z} = \{Z_j | j = 1, ..., n_z\}$. If $0 \in \underline{Z}$ then $KG_h(x) \geq 0$ for all $x \in X$ and if $x$ is sampled infinitely often $KG_h(x) = 0$.*

Proof is in the SM 1. The $Z_j$ values can be fixed, for $n_z = 5$ we use equal Gaussian quantiles $\underline{Z} = \big\{\Phi^{-1}(0.1), \Phi^{-1}(0.3), \ldots, \Phi^{-1}(0.9)\big\}$ where $\Phi(\cdot)$ is the Gaussian CDF. Using quantile spacing and odd $n_z$ ensures $Z_j = \Phi^{-1}(0.5) = 0$ is included which satisfies the assumptions of asymptotic convergence. See Fig.2 (right).

The computational complexity of a single call to ConBO requires the posterior variance ($O(n^2)$) and $n_s n_z$ runs of `Optimizer()`. Let $n_{calls}$ be the number of times that `Optimizer()` calls the posterior mean costing $O(n)$. Thus, ConBO total complexity is $O(n^2 + n n_{calls} n_s n_z)$. Note this is linear in $n_s n_z$, the size of the (small) dynamic optimal discretization over $S \times X$. Thompson sampling with

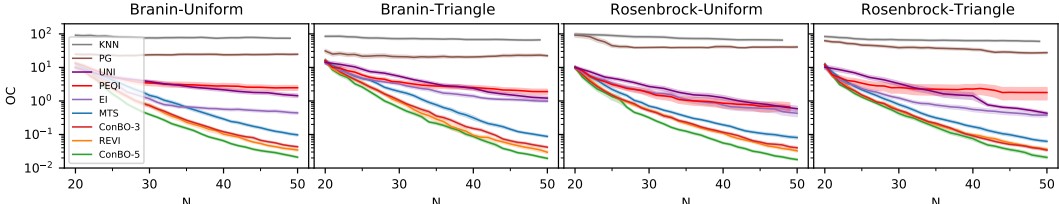

Figure 3: Opportunity Cost across a range of synthetic test problems. The dummy baseline, KNN, is worst in all cases, policy gradient is better however the Gaussian process based methods all perform better. UNI and EI are not conditional algorithms yet outperform PEQI. Amongst other conditional algorihtms, MTS, REVI and ConBO methods all perform significantly better. ConBO-3 is outperformed by ConBO-5 demonstrating the improvement with more accurate $KG_h$.

discretization uses one operation that scales cubically as $O(n^2(n_s n_z) + n(n_s n_z)^2 + (n_s n_z)^3)$ in the worst case and to reduce cost special techniques are required e.g. Fourier features, CG matrix inversion.

# 3 Experiments

We consider synthetic benchmarks and the three applications described in Section 1. In SM 4 we also present parallelization batch sampling results, and experiments with contextual, multi-task and multi-fidelity methods in SM 6. We also briefly test Hybrid KG for global optimization, SM 5, and observe that $KG_{MC}$ performs worse in the same computation time hence we exclude $KG_{MC}$ methods from the conditional experiments. For each benchmark, for evaluation, held out test tasks are sampled from $\mathbb{P}[s] \propto W(s)$, for each test task, $s_i^{test}$, the predicted optimal input is computed, $x^*(s_i^{test})$ and we report the true black box output averaged over all test tasks, see SM 3 for details.

## 3.1 How Accurate is Hybrid Knowledge Gradient?

The theoretical Knowledge Gradient cannot be analytically computed and must be approximated (similarly, entropy based methods are not fully tractable). To illustrate the quality of the approximations, we collected 20 points from the Rosenbrock test function and fit a Gaussian process. Another point was then randomly selected and true KG was estimated by the different methods, keeping the 20 + 1 points fixed and only varying the $n_z$ parameter. Each estimate of KG was recalculated 50 times and Table 1 shows the mean and two standard deviations. Discrete KG results in an extremely loose, volatile bound. The (state of the art) MC method with $n_z = 50$ has a run-to-run variance of $\pm 35\%$, variance reduction techniques used in MC++ (latin hypercube inverse sampling and control variates) can reduce this to $\pm 6\%$. Hybrid KG with $n_z = 3$ is extremely stable, run-to-run variance is too small to show, and is within the error margins of MC++ with $n_z = 50$ which is $\sim 17\times$ more expensive to compute. Increasing $n_z$ tightens the bound, hybrid KG with $n_z = 5$ has 98.2% of the value of $n_z = 50$. As a result, Hybrid KG is much easier to optimize, the Adam optimizer may be used with a much higher learning rate and lower momentum and thus converges much faster.

As an aside, the recently proposed one-shot KG [34] enhances the optimization of $KG_{MC}$. By freezing the $Z$ values between calls to $KG_{MC}$ thus $\tilde{X}^*$ may be reused, this enables *joint* gradient ascent of $(x^{n+1}, \tilde{X}^*)$. In the global optimization use case (a constant single task) one-shot KG and Hybrid KG can be combined. In the conditional setting, the $\tilde{X}_i^*$ of past sampled tasks may be saved, however each call to ConBO must sample new tasks that are not in the saved history, one-shot KG style joint optimization is not possible. In our implementation we utilise caching of $\tilde{X}_i^*$ from old tasks to heuristically warm start finding $\tilde{X}_i^*$ for new tasks, see SM 3.

## 3.2 Synthetic Functions

We perform low-dimensional toy experiments in an ideal setting as a sanity check where we expect all conditional methods to perform similarly. We use the popular Branin-Hoo and Rosenbrock test functions in 2D defining the (task, input) domain as displayed in Figure 1.

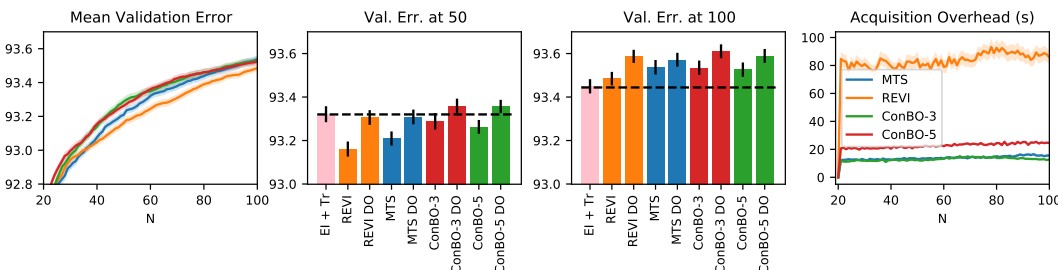

Figure 4: Left: validation accuracy. Centre-left: validation accuracy after 50 samples. Centre-right: validation accuracy after 100 samples. Right: algorithm overhead in seconds. After 50 samples, none of the multi-task models outperform the baseline, EI + Tr (dashed line) suggesting all datasets can use similar hyperparameters. For the larger budget 100, all models outperform the baseline by $0.1\%$ suggesting that for more fine-tuning, each dataset requires different hyperaparameters. In all cases, performing data optimization significantly increases performance.

**Synthetic Functions** Using the Rosenbrock and Branin-Hoo functions, we consider a uniform and a triangular task weighting $W(s)$. For baselines, we adopt two policy based methods. **KNN**: (dummy baseline) randomly collect data, $x^*_{KNN}(s)$ takes a task, $s$, and returns the best input from 10 nearest neighbor tasks. **PG**: policy gradient, a parametric quadratic policy $x^*_{PG}(s) = \pi_\theta(s)$ is learnt by maximising observed performance values, each iteration samples a task from $\mathbb{P}[s] \propto W(s)$ then $x$ is sampled with an $\epsilon$-greedy strategy. For a controlled ablation study, all the BO methods fit the same GP and for evaluation use the same definition of $x^{*N}(s)$ (Equation 3), methods only differ by their data acquisition strategy. **UNI**: random data collection, the most recent conditional methods **PEQI**, **REVI**, **MTS** with all parameters given in SM 3. **ConBO-$n_z$**: given $(s, x)$, 20 tasks are importance sampled, $KG_h$ with $n_z = 3, 5$ points is used. **EI**: expected improvement that treats $(s, x)$ as inputs to be optimized.

Results are shown in Figure 3. Policy based methods KNN and PG consistently perform worse than the Gaussian process methods. Surprisingly, the conditional BO algorithm PEQI performs similarly to UNI and much worse than EI. All other conditional methods outperform all non-conditional methods.

### 3.3 CNN Training Hyperparameters

We apply MTS, REVI, and ConBO variants, and we adopt the recently proposed kernel used for BO with Common Random Numbers [35],

$$k((s, x), (s', x')) = \sigma_0^2 M(x, x'; \underline{l}) + \delta_{s's}(\sigma_1^2 M(x, x'; \underline{l}) + \sigma_3^2).$$

where $M(x, x'; \underline{l})$ is a Matérn $\frac{5}{2}$ kernel with length scales $\underline{l}$. The first term models a common trend function across all tasks and the second term models how each task independently differs from the trend. The differences are composed of another Matérn and the constant kernel to model a global offset e.g. one dataset may have universally higher validation accuracy. This kernel has far fewer parameters than a full multi-task product kernel, it is easy to fit and scales to an arbitrary number of tasks (or datasets) without adding extra parameters.

In this problem setting, learning hyperparameters over similar datasets, one may expect that the optimal hyperparameters would be the same for all datasets. Therefore, as a baseline we apply EI to learn the hyperparameters of the first dataset (task 1). We then evaluate the objective function (validation accuracy) on the rest of the datasets using the best observed hyperparameters from dataset 1, we refer to this as **EI + Tr**ansfer.

**Argument Optimization versus Data Optimization** In continuous task settings, it is not possible to evaluate every task. In discrete task settings with large sampling budgets $N \gg |S|$, a user may desire a single high (although stochastic) output value, $\max y$, for every task. For example, in network hyperparameter optimization, the network with the best validation error, $\max y$, will be deployed (and the hyperparameters $x$ may or may not be reused). We refer to this is *data optimization* (DO). For simulated environments, the input (or "action") that generalizes providing the best long-term average performance, $\max_x \mathbb{E}[f(s, x)]$, is deployed and we refer to this is *argument optimization* (AO). Past work [22] has shown that argument optimization finds inputs that generalize better but

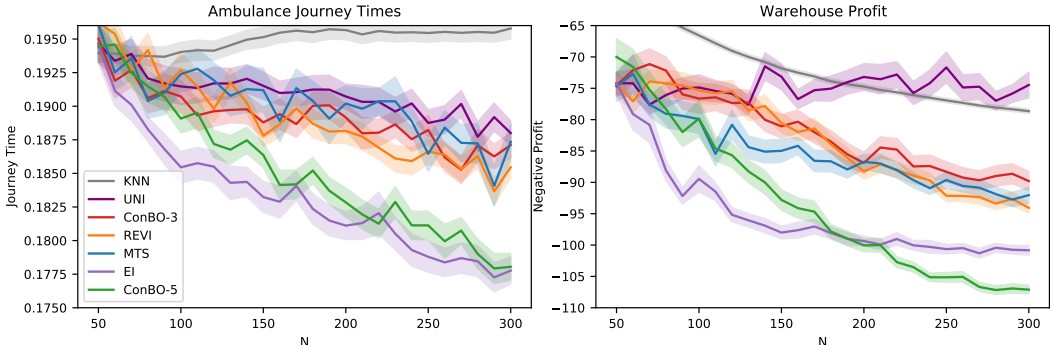

Figure 5: Left: average journey times across a range of cities. Right : average profit across a range of warehouses. ConBO-5 and EI perform best on these benchmarks.

may not provide optimal $\max y$ for all tasks during the optimization run. The authors propose a "DO trick", use an AO method for $N - |S|$ iterations, then finally allocate one sample per task with input, $x^{n+1}$, determined by EI within the task. We apply this trick to all algorithms in this experiment.

Results are shown in Figure 4. For the medium budget of 50 samples, ConBO performs best of the standard algorithms yet it is still worse than the EI+Tr baseline. Applying the final round of DO improves all results to match the baseline. For the large budget of 100 samples, all methods outperform the baseline suggesting dataset specific fine-tuning of hyperparameters is required to achieve best results. Again, DO provides a significant boost to performance for all methods.

Gaussian process kernel parameter learning required approximately 2–5 seconds using Tensorflow. In Figure 4 (right) we show the runtime of each algorithm excluding model fitting and network training, purely acquisition function optimization time. MTS and ConBO-3 are quickest while Conbo-5 increases linearly over ConBO-3 and REVI takes much longer.

### 3.4 Ambulances and Warehouses

We apply all methods from Section 3.2 to two benchmarks from the `www.SimOpt.org` library for simulation optimization problems. The ambulance problem (AMB) is 8-dimensional and consists of a range of cities and one must optimize ambulance locations for each city. The Assemble-to-order problem (ATO) is 9-dimensional consisting of a range of warehouses and one must optimize target stock level for each warehouse. Results are shown in Figure 5. Of the policy based methods, PG performs poorly and does not show on the plots whilst KNN performs poorly on AMB and performs well on ATO suggesting that AMB is a more difficult problem. Of the GP based methods, EI performs well for smaller budgets. Although it is not a conditional algorithm we include it to highlight that sometimes the simplest idea can also work. Of the conditional methods, MTS, REVI, and ConBO-3 all perform similarly, either slightly (AMB) or largely (ATO) outperforming UNI. These methods struggle in higher dimensions while ConBO-5 uses a more accurate acquisition function and is the only method that consistently performs well *across all problems*. We hypothesize that these problems are more difficult than the synthetics and CNN and truly stress test conditional algorithms.

## 4 Conclusion

**Potential Limitations and Broader Impact** there are multiple ways in which ConBO may fail, in this work we have not investigated how ConBO or Hybrid KG suffers with poorly learnt Gaussian process hyperparameters. In many applications, a poorly chosen kernel or unoptimized hyperparameters can lead to poor performance and our proposed methods may be more sensitive or more robust to these failures than alternative approaches. We propose a general purpose optimization algorithm and analysis, we do not currently see any immediate societal impact.

We investigate Conditional Bayesian optimization and propose ConBO. ConBO is designed from the ground up to fully exploit the structure of conditional problems, namely that optimizing one task helps optimize similar tasks. Hence every point should be collected to maximise the benefit of all tasks. However, this can lead to excessive computational cost, particularly in higher dimensions. Thus we also propose Hybrid KG that mixes past methods to be both fast and scalable.

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
