# Practical Bayesian Optimization of Objectives with Conditioning Variables: Supplementary Material

Annonymous

February 2021

## 1 Theoretical Results

We first state the standard assumption in Bayesian Optimization. We then restate the theorems from the main paper and provide each proof. Firstly, in Theorem 1 we then show that that ConBO with knowledge gradient myopically maximises *Value of Information* in a Bayesian decision theoretic framework. In theorem 2 we show that in discrete settings ConBO will sample all pairs infinite often. Finally, in Theorem 3 we prove conditions for hybrid KG to satisfy the results of Theorems 1 and 2.

**Assumption 1** *Let $\theta(s,x) = \mathbb{E}[f(s,x)]$ be the latent function of expected performance (average over output noise). Let $k((s,x),(s'x'))$ be the kernel of a Gaussian process. We assume that $\theta(s,x)$ is in the Reproducing Kernel Hilbert Space of the kernel $k((s,x),(s'x'))$.*

**Theorem 1** *Let $(s^*,x^*) \in \arg\max ConBO(s,x)$ be a point chosen for sampling. $(s^*,x^*)$ is also the point that maximises the myopic Value of Information, the increase in predicted performance.*

*Proof of Theorem 1* Given all the information available at time $n$, $\tilde{X}^n, Y^n$ and the model $\mu^n(s,s)$, $k^n(s,x,s'x')$, for any given task and input $s,x$, and a given realization of the true reward function $f()$, in a Bayesian decision theoretic framework, the loss is given by the output of the function

$$\text{Loss}(s,x) = -f(s,x),$$

the expected loss is the risk function

$$\text{Risk}(s,x) = \mathbb{E}[\text{Loss}(s,x)|\tilde{X}^n, Y^n] = \mathbb{E}[-f(s,x)|\tilde{X}^n, Y^n] = -\mu^n(s,x).$$

For convenience we assume conditioning on $\tilde{X}^n, Y^n$ for the remaining equations. The optimal input minimizes risk

$$x^{optimal} = \arg\min_x \text{Risk}(s,x) = \arg\min_x -\mu^n(s,x).$$

Alternatively, $x^{*n}(s) = \arg\max_x \mu^n(s,x)$ is the Bayesian decision theoretic optimal input given all data available at time $n$. The total risk is the risk of optimal inputs is for all tasks, or the risk of the chosen inputs

$$\text{Total Risk}(n) = -\int_s \max_x \mu^n(s,x)\mathbb{P}[s]ds$$

which is the negative of the models best prediction of true reward given data up to time $n$ where we have made $n$ an explicit argument for convenience. Next assume we are able to collect more data to update the model, choose $(s,x)^{n+1}$ and observe $y^{n+1}$. The myopic *Value of Information* is defined as the data that minimizes future risk

$$\text{VoI}((s,x)^{n+1}) = -\mathbb{E}_{y^{n+1}}[\text{Total Risk}(n+1)|(s,x)^{n+1}] \tag{1}$$

$$\tag{2}$$

Note that that $\arg\max \mathrm{VoI}((s,x)^{n+1})$ is not affected by adding terms that do not depend on $(s,x)^{n+1}$. Thus we may subtract the current Total Risk$(n)$. Finally, the difference between risks simplifies to

$$\mathbb{E}_{y^{n+1}}\left[\int_s \max_x \mu^{n+1}(s,x)\mathbb{P}[s]ds\,\Big|\,(s,x)^{n+1}\right] - \int_{s'} \max_x \mu^n(s',x)\mathbb{P}[s]ds' \tag{3}$$

$$= \mathbb{E}_{y^{n+1}}\left[\int_s \max_x \mu^{n+1}(s,x) - \max_x \mu^n(s,x)\mathbb{P}[s]ds\,\Big|\,(s,x)^{n+1}\right] \tag{4}$$

$$= \int_s \mathbb{E}_{y^{n+1}}\left[\max_x \mu^{n+1}(s,x)|(s,x)^{n+1}\right] - \max_x \mu^n(s,x)\mathbb{P}[s]ds \tag{5}$$

$$= \int_s \mathrm{KG}_c(s;(s,x)^{n+1})\mathbb{P}[s]ds \tag{6}$$

$$= \mathrm{ConBO}((s,x)^{n+1}) \tag{7}$$

Therefore $\arg\max \mathrm{VoI}((s,x)^{n+1}) = \arg\max \mathrm{ConBO}((s,x)^{n+1})$. $\square$

**Theorem 2** *Let $S$ and $X$ be finite sets and $N$ the budget to be sequentially allocated by ConBO. Let $n(s,x,N)$ be the number of samples allocated to point $s,x$ within budget $N$. Then for all $(s,x) \in S \times X$ we have that $\lim_{N\to\infty} n(s,x,N) = \infty$.*

We require some intermediate results, firstly ConBO is non-negative.

**Lemma 1** *Let $(s,x) \in S \times X$, then $ConBO((s,x)^{n+1}) \geq 0$.*

*Proof of Lemma 1*

$$\mathrm{ConBO}((s,x)^{n+1}) = \sum_{s'} \mathbb{E}_Z[\max_{x'} \mu^n(s',x') + \tilde{\sigma}(s',x';(s,x)^{n+1})Z] - \max_{x''} \mu^n(s',x'') \tag{8}$$

$$\geq \sum_{s'} \mathbb{E}_Z[\mu^n(s',\pi^n(s')) + \tilde{\sigma}(s',\pi^n(s');(s,x)^{n+1})Z] - \max_{x''} \mu^n(s',x'') \tag{9}$$

$$= \sum_{s'} \max_{x'} \mu^n(s',x') + \tilde{\sigma}(s',\pi^n(s');(s,x)^{n+1})\mathbb{E}_Z[Z] - \max_{x''} \mu^n(s',x'') \tag{10}$$

$$= \sum_{s'} \max_{x'} \mu^n(s',x') - \max_{x''} \mu^n(s',x'') \tag{11}$$

$$= 0 \tag{12}$$

$\square$

Secondly, we require that $\mathrm{ConBO}(s,x)$ reduces to zero for an infinitely sampled pair. Note that for deterministic $f(s,x)$, the result simplifies to $\mathrm{ConBO}(s,x)$ is zero for any sampled pair.

**Lemma 2** *Let $(s,x)^{n+1} \in S \times X$ with $n(s,x) = \infty$, then $ConBO((s,x)^{n+1}) = 0$.*

*Proof of Lemma 2* Given infinitely many finite variance observations of $f(s,x)$, we have that $\mu^n(s,x) = \mathbb{E}[f(s,x)]$ and posterior variance is zero $k^n(s,x,s,x) = 0$. By the positive definiteness of the kernel we also have that $k^n(s,x,s',x') = 0$ for all $(s',x')^{n+1} \in S \times X$ (see [1] Lemma 3). It follows that $\tilde{\sigma}(s',x';(s,x)^{n+1}) = 0$ for all $(s',x') \in S \times X$ and thus

$$\mathrm{KG}_c(s';(s,x)^{n+1}) = \mathbb{E}_Z[\max_{x'} \mu^n(s',x') + \tilde{\sigma}(s',x';(s,x)^{n+1})Z] - \max_{x''} \mu^n(s',x'') \tag{13}$$

$$= \mathbb{E}_Z[\max_{x'} \mu^n(s,x') + 0 \cdot Z] - \max_{x''} \mu^n(s',x'') \tag{14}$$

$$= \max_{x'} \mu^n(s',x') - \max_{x''} \mu^n(s',x'') \tag{15}$$

$$= 0 \tag{16}$$

and therefore $\text{ConBO}((s,x)^{n+1}) = \int_s 0\, \mathbb{P}[s]ds = 0$.
$\square$

Thirdly, we require the inverse of Lemma 2, that points for which $\text{ConBO}(s,x) > 0$ must have non-zero variance $k^n(s,x,s,x) > 0$ (and therefore cannot be infinitely sampled).

**Lemma 3** *Let $(s,x)^{n+1} \in S \times X$ be a point for which $\text{ConBO}((s,x)^{n+1}) > 0$, then $n(s,x) < \infty$.*

*Proof of Lemma 3* $\text{ConBO}((x,s)^{n+1}) > 0$ implies that there exists an $s \in S'$ such that $KG_c(s';(s,x)^{n+1}) > 0$. By the contrapositive of Lemma 3 in [2], we must have that $k^n(s',x',(s,x)^{n+1})$ is *not* a constant function of $x'$. If $(s,x)^{n+1}$ is infinitely sampled, then $k^n(s',x',(s,x)^{n+1})$ is a constant function of $x'$, thus $(s,x)^{n+1}$ is not infinitely sampled. $\square$

Finally, combining the previous Lemmas we can complete the proof.

*Proof of Theorem 2* By Lemmas 1 and 2, any infinitely sampled points become minima of the function $\text{ConBO}(s,x)$. By construction, the ConBO algorithm choose points at maxima $(s,x)^{n+1} = \arg\max \text{ConBO}(s,x)$. Thus in the infinite budget limit, we have $\text{ConBO}(s,x) = 0$ for all $(s,x) \in S \times X$ by the contrapositive of Lemma 3 we have that $n(s,x) = \infty$ for all points.
$\square$

**Theorem 3** *Let $n_z \geq 2$ and let $\underline{Z} = \{Z_j | j = 1,...,n_z\}$. If $0 \in \underline{Z}$ then $KG_h(x) \geq 0$ for all $x \in X$ and if $x$ is sampled infinitely often $KG_h(x) = 0$.*

*Proof of Theorem 3* First consider the base case $n_z = 2$. Let $\underline{Z} = \{0, Z_2\}$ and given $x^{n+1}$, let $X^* = \{x_1^*, x_2^*\}$ be the optimal discretization as found by Algorithm 3. Then $x_1^* = \arg\max \mu^n(x) + \tilde{\sigma}(x; x^{n+1}) \cdot 0 = \arg\max \mu^n(x)$ and therefore $\mu^n(x_1^*) = \max_x \mu^n(x)$. Let $\mu^* = \mu^n(X^*)$ and $\tilde{\sigma}^* = \tilde{\sigma}(X^*, x^{n+1})$. Then we have that

$$
\begin{align}
KG_h(x^{n+1}) &= \mathbb{E}_Z\left[\max\{\mu_1^* + \tilde{\sigma}_1^* Z, \mu_2^* + \tilde{\sigma}_2^* Z\}\right] - \max_x \mu^n(x) \tag{17}\\
&= \mathbb{E}_Z\left[\max\{\max_x \mu^n(x) + \tilde{\sigma}_1^* Z, \mu_2^* + \tilde{\sigma}_2^* Z\}\right] - \max_x \mu^n(x) \tag{18}\\
&= \mathbb{E}_Z\left[\max\{\tilde{\sigma}_1^* Z, \mu_2^* - \max_x \mu^n(x) + \tilde{\sigma}_2^* Z\}\right] \tag{19}\\
&\geq \max\left\{\mathbb{E}_Z\left[\tilde{\sigma}_1^* Z\right], \mathbb{E}_Z\left[\mu_2^* - \max_x \mu^n(x) + \tilde{\sigma}_2^* Z\right]\right\} \tag{20}\\
&= \max\left\{0, \mu_2^* - \max_x \mu^n(x)\right\} \tag{21}\\
&= 0 \tag{22}
\end{align}
$$

where we Jensen's inequality in the penultimate line and we use that $\mu_2^* < \max_x \mu^n(x)$ in the final line. The result extends to the case for $n_z > 2$ trivially. The proof for $KG_h(x) = 0$ at infinitely sampled points follows the proof of Lemma 2. $\square$

# 2 Computing ConBO and Hybrid Knowledge Gradient

## 2.1 Deriving One-Step Look-ahead Posterior Mean $\mu^{n+1}(s,x)$

At iteration $n$ during optimization, let the training inputs be $\tilde{X}^n = ((s^1,x^1),...,(s^n,x^n))$ and the training outputs $Y^n = (y^1,...,y^n)$. Given a prior mean and kernels functions, $\mu^0(s,x): S \times X \to \mathbb{R}$ and $k^0(s,x,s',x'): S \times X \times S \times X \to \mathbb{R}$. Finally let the new sample point be $(s,x)^{n+1} = \tilde{x}^{n+1}$.

Updating the mean function with data from the $0^{th}$ step to $n^{th}$ step is given by

$$
\begin{align}
\mu^n(s,x) &= \mu^0(s,x) + k^0(s,x,\tilde{X}^n)\underbrace{K^{-1}\left(Y^n - \mu^0(\tilde{X}^n)\right)}_{\text{define } \tilde{Y}^n} \tag{23}\\
&= \mu^0(s,x) + k^0(s,x,\tilde{X}^n)\tilde{Y}^n \tag{24}
\end{align}
$$

**Algorithm 1** The ConBO algorithm. At each iteration a task and an input are chosen and evaluated. We use the shorthand $\tilde{X}^n$ to represent all sampled $(s, x)$ pairs. The Adam optimizer is used to maximize $\text{ConBO}(s, x)$.

---

**Require:** Problem setting: $f : S \times X \to \mathbb{R}$, $W(s)$, $N$
**Require:** Algorithm parameters: $n_0$, $n_s$, $n_z$, kernel
    Initialize dataset $\tilde{X}^{n_0}$ and $Y^{n_0}$
    **for** $n$ in $n_0, \ldots, N$ **do**
        Fit the model $\mu^n(\cdot), k^n(\cdot) \leftarrow \text{GP}(\tilde{X}^n, Y^n, \text{kernel})$
        Acquire point $(s, x)^{n+1} \leftarrow \text{Adam}\,(\text{ConBO}(s, x))$
        Evaluate point $y^{n+1} \leftarrow f(s^{n+1}, x^{n+1})$
        Update dataset $Y^{n+1}$ and $\tilde{X}^{n+1}$
    **end for**
    Update the model $\mu^N(s, x) \leftarrow \text{GP}(\tilde{X}^N, Y^N, \text{kernel})$
    **return** conditional maxima function, $x^{*N}(s) = \arg\max_x \mu^N(s, x)$

---

**Algorithm 2** Computing ConBO. The algorithm requires a new point, discretization sizes, past data and posterior GP functions and an optimizer.

---

**Require:** $\tilde{x}^{n+1}$, $n_s$, $n_z$, $\tilde{X}^n$, $Y^n$, $\mu^n(s, x)$, $k^n(s, x, s'x')$, $\sigma_\epsilon^2$, `Optimizer()`
    Precompute and cache $\left(k^0(\tilde{X}^n, \tilde{X}^n) + \sigma_\epsilon^2 I\right)^{-1} k^0(\tilde{X}^n, \tilde{x}^{n+1})$
    $C \leftarrow 0$
    **for** $i$ in $1, .., n_s$ **do**
        $s_i \sim N(s_i | s^{n+1}, \text{diag}(l_s^2))$
        $KG_i \leftarrow \text{Algorithm3}(\tilde{x}^{n+1}, s_i, ....)$
        $w_i \leftarrow \mathbb{P}[s_i] / N(s_i | s^{n+1}, \text{diag}(l_s^2))$
        $C \leftarrow C + w_i KG_i / n_s$
    **end for**
    **return** $C$

---

where $K = k^0(\tilde{X}^n, \tilde{X}^n) + \sigma_\epsilon^2 I$. $\mu^n(s, x)$ may also be written as a weighted average of a modified $\tilde{Y}^n \in \mathbb{R}^n$ vector as defined above. Computing the new posterior mean reduces to augmenting $\tilde{X}^n \to \tilde{X}^{n+1}$ with $\tilde{x}^{n+1}$ and $Y^n \to Y^{n+1}$ then computing the new $\tilde{Y}^{n+1} \in \mathbb{R}^{n+1}$. Let $Z$ be the z-score of $y^{n+1}$ on its predictive distribution, then

$$\tilde{Y}^{n+1} = \begin{bmatrix} \tilde{Y}^n \\ 0 \end{bmatrix} + \frac{Z}{\sqrt{k^n(\tilde{x}^{n+1}, \tilde{x}^{n+1}) + \sigma_\epsilon^2}} \begin{bmatrix} -K^{-1} k^0(\tilde{X}^n, \tilde{x}^{n+1}) \\ 1 \end{bmatrix} \tag{25}$$

and the above expression may be used directly in Algorithm 1 with sampled $Z_j \sim N(0, 1)$. This is derived by a simple change of indices from $0 \to n$ and $n \to n+1$, yields the one-step updated posterior mean

$$\mu^{n+1}(s, x) = \mu^n(s, x) + \frac{k^n(s, x, \tilde{x}^{n+1})}{k^n(\tilde{x}^{n+1}, \tilde{x}^{n+1}) + \sigma_\epsilon^2} \left(y^{n+1} - \mu^n(\tilde{x}^{n+1})\right). \tag{26}$$

which contains the random $y^{n+1}$. This may be factorized as follows:

$$\mu^{n+1}(s, x) = \mu^n(s, x) + k^n(s, x, \tilde{x}^{n+1}) \underbrace{\frac{1}{\sqrt{k^n(\tilde{x}^{n+1}, \tilde{x}^{n+1}) + \sigma_\epsilon^2}}}_{\text{standard deviation of } y^{n+1}} \underbrace{\frac{\left(y^{n+1} - \mu^n(\tilde{x}^{n+1})\right)}{\sqrt{k^n(\tilde{x}^{n+1}, \tilde{x}^{n+1}) + \sigma_\epsilon^2}}}_{\text{Z-score of } y^{n+1}} \tag{27}$$

$$= \mu^n(s, x) + k^n(s, x, \tilde{x}^{n+1}) \frac{1}{\sigma_{y^{n+1}}^n(\tilde{x}^{n+1})} Z \tag{28}$$

$$= \mu^n(s, x) + \tilde{\sigma}(s, x, \tilde{x}^{n+1}) Z \tag{29}$$

**Algorithm 3** Computing Hybrid Knowledge Gradient. The algorithm requires a new point, a task, a discretization size, past data, posterior GP functions and an optimizer.

**Require:** $\tilde{x}^{n+1}$, $s$, $n_z$, $\tilde{X}^n$, $Y^n$, $\mu^n(s,x)$, $k^n(s,x,s'x')$, $\sigma_\epsilon^2$, `Optimizer()`

$\quad \tilde{X}^{n+1} \leftarrow \tilde{X}^n \cup \{\tilde{x}^{n+1}\}$

$\quad X^* \leftarrow \{\}$

$\quad$ Precompute and cache $\left(k^0(\tilde{X}^n, \tilde{X}^n) + \sigma_\epsilon^2 I\right)^{-1} k^0(\tilde{X}^n, \tilde{x}^{n+1})$

$\quad$ **for** $j$ **in** $1, .., n_z$ **do**

$\quad\quad Z_j \leftarrow \Phi^{-1}\left(\frac{2j-1}{2n_z}\right)$

$\quad\quad \tilde{Y}_j^{n+1}$ from Equation 25

$\quad\quad \mu_j^{n+1}(s,x) \leftarrow \mu^0(s,x) + k^0(s,x,\tilde{X}^{n+1})\tilde{Y}_j^{n+1}$

$\quad\quad x_j^* \leftarrow \arg\max_x \mu_j^{n+1}(s,x)$ using `Optimizer()`

$\quad\quad X^* \leftarrow X^* \cup \{x_j^*\}$

$\quad$ **end for**

$\quad \underline{\mu} \leftarrow \mu^n(s, X^*)$

$\quad \underline{\sigma} \leftarrow \frac{k^n((s,X^*),\tilde{x}^{n+1})}{\sqrt{k^n(x^{n+1},x^{n+1})+\sigma_\epsilon^2}}$

$\quad$ KG $\leftarrow$ Algorithm4$(\underline{\mu}, \underline{\sigma})$

$\quad$ **return** KG

---

**Algorithm 4** Knowledge Gradient by discretization. This algorithm takes as input a set of linear functions parameterised by a vector of intercepts $\underline{\mu}$ and a vector of gradients $\underline{\sigma}$. It then computes the intersections of the piece-wise linear epigraph (ceiling) of the functions and the expectation of the output of the function given Gaussian input. Vector indices are assumed to start from 0.

**Require:** $\underline{\mu}, \underline{\sigma} \in \mathbb{R}^{n_A}$

$\quad O \leftarrow \text{order}(\underline{\sigma})$ $\quad\quad$ # get sorting indices of increasing $\underline{\sigma}$

$\quad \underline{\mu} \leftarrow \underline{\mu}[O], \underline{\sigma} \leftarrow \underline{\sigma}[O]$ $\quad$ # arrange elements

$\quad I \leftarrow [0, 1]$ $\quad\quad\quad\quad$ # indices of elements in the epigraph

$\quad \tilde{Z} \leftarrow [-\infty, \frac{\mu_0-\mu_1}{\sigma_1-\sigma_0}]$ $\quad$ # z-scores of intersections on the epigraph

$\quad$ **for** $i = 2$ **to** $n_z - 1$ **do**

$\quad\quad (\star)$

$\quad\quad j \leftarrow last(I)$

$\quad\quad z \leftarrow \frac{\mu_i-\mu_j}{\sigma_j-\sigma_i}$

$\quad\quad$ **if** $z < last(\tilde{Z})$ **then**

$\quad\quad\quad$ Delete last element of $I$ and of $\tilde{Z}$

$\quad\quad\quad$ Return to $(\star)$

$\quad\quad$ **end if**

$\quad\quad$ Add $i$ to end of $I$ and $z$ to $\tilde{Z}$

$\quad$ **end for**

$\quad \tilde{Z} \leftarrow [\tilde{Z}, \infty]$

$\quad \underline{A} \leftarrow \phi(\tilde{Z}[1:]) - \phi(\tilde{Z}[:-1])$ $\quad\quad$ # assuming python indexing

$\quad \underline{B} \leftarrow \Phi(\tilde{Z}[1:]) - \Phi(\tilde{Z}[:-1])$

$\quad$ KG $\leftarrow \underline{B}^T\underline{\mu}[I] - \underline{A}^T\underline{\sigma}[I] - \max\underline{\mu}$ $\quad$ # compute expectation

$\quad$ **return** KG

---

where the left factor is a deterministic and the right factor is the (at time $n$) stochastic Z-score of the new $y^{n+1}$ value. This is clear by noting that the predictive distribution of the new output $y^{n+1}$

$$\mathbb{P}[y^{n+1}|\tilde{x}^{n+1}, \tilde{X}^n, Y^n] = N(\mu^n(\tilde{x}^{n+1}), k^n(\tilde{x}^{n+1}, \tilde{x}^{n+1}) + \sigma_\epsilon^2). \tag{30}$$

as a result, to sample new posterior mean functions, we may simply sample $Z \sim N(0,1)$ values and compute Equation 28. However, this results in a quadratic cost per call to sampled poster mean function as both $k^n(s,x,\tilde{x}^{n+1})$ and $\sigma^n_{y^{n+1}}(\tilde{x}^{n+1})$ have $O(n^3)$quadratic cost. This can be easily reduced to linear instead as we now show.

We next focus on the first factor $k^n(s,c,\tilde{x}^{n+1})$ which may also be factorized

$$k^n(s,x,\tilde{x}^{n+1}) = k^0(s,x,\tilde{x}^{n+1}) - k^0(s,x,\tilde{X}^n)K^{-1}k^0(\tilde{X}^n,\tilde{x}^{n+1}) \tag{31}$$

$$= \underbrace{\left[k^0(s,x,\tilde{X}^n), k^0(s,x,\tilde{x}^{n+1})\right]}_{k^0(s,x,\tilde{X}^{n+1})} \begin{bmatrix} -K^{-1}k^0(\tilde{X}^n,\tilde{x}^{n+1}) \\ 1 \end{bmatrix}. \tag{32}$$

Combining Equations 24 and 32 yields the following formula

$$\mu^{n+1}(s,x) = \mu^0(s,x) + k^0(s,x,\tilde{X}^{n+1})\left(\begin{bmatrix} \tilde{Y}^n \\ 0 \end{bmatrix} + \frac{Z}{\sigma^n_{y^{n+1}}(\tilde{x}^{n+1})}\begin{bmatrix} -K^{-1}k^0(\tilde{X}^n,\tilde{x}^{n+1}) \\ 1 \end{bmatrix}\right) \tag{33}$$

$$= \mu^0(s,x) + k^0(s,x,\tilde{X}^{n+1})\tilde{Y}^{n+1}. \tag{34}$$

The quantity $\tilde{Y}^n$ is pre-computed at the start of the algorithm iteration, the quantity $-K^{-1}k^0(\tilde{X}^n,\tilde{x}^{n+1})$ has quadratic cost and can be computed once and used again for $\sigma^n_{y^{n+1}}(\tilde{x}^{n+1})$. Then, sampling posterior mean functions reduces to sampling $n_y$ values $z_1,...,z_{n_y} \sim N(0,1)$ and for each value computing $\tilde{Y}^{n+1}_1,....,\tilde{Y}^{n+1}_{n_y}$. Then each sampled posterior mean is just the weighted average given by Equation 34.

## 2.2  Choice of `Optimizer()`

Since evaluating Equation 34 for many points is simply a matrix multiplication, random search is cheap to evaluate in parallel. After random search, the gradient of Equation 34 with respect to $(s,x)$ is easily computed, and starting from the best random search point, gradient ascent over $x \in X$ can be used to find the optimal input. This varies with kernel choice and application, we describe our settings in Section 3.4.

We start with a set of sampled means $\mu^{n+1}_1(s,x),....,\mu^{n+1}_{n_z}(s,x)$ and a set of sampled tasks $s_1,...,s_{n_s}$. For each task $s_i$, the set of $n_z$ optimal inputs $X_{d,s_i}$ is found by optimizing the $n_z$ posterior means

$$X_{d,s_i} = \bigcup_{j=1}^{n_z}\left\{\operatorname*{argmax}_x \mu^{n+1}_j(s_i,x)\right\}$$

Finally, for each point in the $\{s_i\} \times X_{d,s_i} = \tilde{X}_{d,s_i}$ (task, input) discretization, we evaluate two quantities, firstly the vector of current posterior means $\underline{\mu}_{s_i} \in \mathbb{R}^{n_z}$,

$$\underline{\mu}_{s_i} = \mu^n(\tilde{X}_{d,s_i}) \tag{35}$$

$$= \mu^0(\tilde{X}_{d,s_i}) + k^0(\tilde{X}_{d,s_i},\tilde{X}^n)\tilde{Y}^n \tag{36}$$

and the vector of additive updates $\underline{\sigma}_{s_i} \in \mathbb{R}^{n_z}$,

$$\underline{\sigma}_{s_i} = \frac{k^n(\tilde{X}_{d,s_i},\tilde{x}^{n+1})}{\sigma^n_{y^{n+1}}(\tilde{x}^{n+1})} \tag{37}$$

$$= k^0(\tilde{X}_{d,s_i},\tilde{X}^{n+1})\begin{bmatrix} -K^{-1}k^0(\tilde{X}^n,\tilde{x}^{n+1}) \\ 1 \end{bmatrix}\frac{1}{\sigma^n_{y^{n+1}}(\tilde{x}^{n+1})}. \tag{38}$$

These two vectors $\underline{\mu}_{s_i}$ and $\underline{\sigma}_{s_i}$ are both differentiable $\nabla_{\tilde{x}^{n+1}}\underline{\mu}_{s_i}$ and $\nabla_{\tilde{x}^{n+1}}\underline{\sigma}_{s_i}$ and they are used to analytically compute the peicewise-linear $\mathbb{E}_Z\left[\max\left(\underline{\mu}_{s_i} + Z\underline{\sigma}_{s_i}\right)\right] - \max\underline{\mu}_{s_i}$ which is also differentiable. Thus assuming fixed $\tilde{X}_{d,s_1},...,\tilde{X}_{d,s_{n_s}}$, approximate gradients are computed and can be used in any stochastic gradient ascent optimizer.

# 3 Experiment Implementation Details

## 3.1 Performance Measurement

We measure convergence of each benchmark by sampling a set of test tasks $S_{test} \sim \mathbb{P}[s] \propto W(s)$ which are never used during optimization. For each optimizer at each iteration we measure the true performance by

$$\text{True Performance} = \frac{1}{N_{test}} \sum_{s_i \in S_{test}} \mathbb{E}[f(s_i, x^{*n}(s_i))].$$

To estimate $\mathbb{E}[f(s, x)]$, for the synthetic functions we simply use the noise free true function

$$\mathbb{E}[f(s, x)] = BraninHoo((s, x))$$

and

$$\mathbb{E}[f(s, x)] = Rosenbrock((s, x)).$$

For the simulators ATO and AMB, these are more expensive to call hence we only evaluate the performance for 15 of the iterations between 20 and 300. This is evaluated by Monte-Carlo with multiple repeated seeds

$$\mathbb{E}[f(s, x)] \approx \frac{1}{N} \sum_{seed} ATO((s, x), seed).$$

## 3.2 REVI

At iteration $n$ of the algorithm, we used a discretization of size $n_{disc} = 2n$, split equally amongst inputs and tasks $n_s = n_x = \lceil \sqrt{n_{disc}} \rceil$. Tasks are sampled from $\mathbb{P}[s] \propto W(s)$ and inputs are sampled as a latin hypercube over $X$. The acquisition function is optimized by 100 points of random search over $S \times X$ followed by Nelder-Mead ascent starting form the best 20 points in the random search phase.

## 3.3 MTS

We use a target discretization size of $n_{disc} = 3000$. Given $d_s$ tasks dimensions and $d_x$ input dimensions, we sampled tasks uniformly, the number of sampled tasks is given by $n_s = \lceil (n_{disc})^{d_s/(d_s+d_x)} \rceil$ and the number of inputs per task is $n_x = \lceil n_{disc}/n_s \rceil$ such that $n_s * n_x \approx n_{disc}$. This way the discretization over all tasks and input dimensions is roughly constant. For each sampled task $s_i$, $n_x$ inputs are generated in three ways. Firstly, the optimal input is evaluated $x^{\pi}_{s_i} = \pi^n(s_i)$, we generate 40 inputs around this optimal input. Secondly, we take the 10 nearest neighbor tasks from the training set, and the points with the 4 largest $y$ values are added to the discretization set with randomly generated neighbors. Finally, remaining inputs in the $n_x$ budget come from uniform random sampling over $X$. Each $s_i$ has a bespoke input discretization. Sampled functions are drawn using the python numpy random normal generate function.

## 3.4 ConBO

Each sampled posterior mean function was optimized in two steps. For a given task $s_i$, firstly, the input discretization used by MTS, reduced to 40 points in total was used in parallel random search. The best point was then used in conjugate-gradient ascent for 20 steps.

For optimizing sampled posterior mean functions for which $z_i = 0$, that is $\mu^{n+1}(s, x) = \mu^n(s, x)$, this given by the optimal input $x^{*n}(s_i)$. Since the same, or very similar tasks, may be used multiple times for different $\text{ConBO}((s, x)^{n+1})$ calls, we may use caching to avoid such repeated optimal input computation. Whenever the optimal input is queried for the optimal input for a given task, the final (task, input) pair are stored in a lookup table. Any future calls to the optimal input function with task $s_j$ can check the lookup table and if very similar tasks exists use the same input, if a somewhat similar task exists, re-optimize the input, if no similar tasks exist perform a full optimization as above.

In our experiments, the cache of stored optimal inputs is wiped clean before any testing, ConBO is not given an unfair advantage at test time. In practical applications, this need not be the case.

## 3.5 Policy Gradient

The (stochastic) policy is a function of task $\pi_\theta(s) = x^*_{PG}(s)$ returning a Gaussian mean and constant variance over input space. The mean is a quadratic function of the task and constant variance,

$$\pi_\theta(s) = \mathbb{P}_\theta[x|s] = N(x|s^T A s + Bs + C, 0.2^2 I)$$

where $A \in \mathbb{R}^{d_X \times d_S \times d_S}$, $B \in \mathbb{R}^{d_X \times d_S}$ and $C \in \mathbb{R}^{d_X}$ and $\theta = \{A, B, C\}$. Given a dataset $\{(s, x, y)^i\}_{i=1}^n$, we first rescaled $s$ and $x$ values to the hypercube, and y-values were standardized to have mean 0 and variance 1. First we used kernel regression to learn a baseline value

$$V(s) = \frac{\sum_i k(s, s^i) y^i}{\sum_i k(s, s^i)}$$

where $k(s, s') = \exp(-0.5(s - s')^2/0.2^2)$. The parameters $\theta$ are found by optimizing the expected advantage

$$\text{Expected advantage} = \sum_i \mathbb{P}_\theta[x^i|s^i](y^i - V(s^i)).$$

At test time, given a task, the mean input is computed from the policy (accounting for rescaling to hypercube and back) and recommended for use.

## 3.6 Cifar10 Hyperparameter Experiment

**Parameter Space:**

- `dropout_1` $\in [0, 0.8]$, linear scale

- `dropout_2` $\in [0, 0.8]$, linear scale

- `dropout_3` $\in [0, 0.8]$, linear scale

- `learning_rate` $\in [0.0001, 0.01]$, log scale

- `beta_1` $\in [0.7, 0.99]$, log scale

- `beta_2` $\in [0.9, 0.999]$, log scale

- `batch_size` $\in [16, 512]$, log scale

**Network architecture:**

```
x_in = Input(shape=(32, 32, 3))

x = Conv2D(filters=64, kernel_size=2, padding='same', use_bias=False)(x_in)
x = BatchNormalization()(x)
x = Activation("relu")(x)
x = MaxPooling2D(pool_size=2)(x)
x = Dropout(dropout_1)(x)

x = Conv2D(filters=32, kernel_size=2, padding='same', use_bias=False)(x)
x = BatchNormalization()(x)
x = Activation("relu")(x)
x = MaxPooling2D(pool_size=2)(x)
x = Dropout(dropout_2)(x)

x = Flatten()(x)
```

```
x = Dense(256, activation='relu')(x)
x = Dropout(dropout_3)(x)
x_out = Dense(2, activation='softmax')(x)

cnn = Model(inputs=x_in, outputs=x_out)

adam = optimizers.Adam(learning_rate=learning_rate,
                       beta_1=beta_1,
                       beta_2=beta_2)
cnn.compile(loss='categorical_crossentropy',
            optimizer=adam,
            metrics=['accuracy'])
```

# 4   Batch Construction by Sequential Penalization

For global optimization, parallelizing BO algorithms to sugggest a batch of $q$ inputs, $\{x^{n+1}, ..., x^{n+q}\}$, has been approached in multiple ways. For acquisitions functions that compute an expectation over future outcomes $\mathbb{P}[y^{n+1}|x^{n+1}]$, (EI, KG, ES, MES), the acquisition value of a batch can be computed using the expectation over multiple correlated outcomes $\mathbb{P}[y^{n+1}, ..., y^{n+q}|x^{n+1}, ..., x^{n+q}]$. This larger $q$ dimensional expectation, effectively looking $q$ steps into the future, must be estimated by Monte-Carlo. At the same time, it is a function of all $q$ points in the batch and must be optimized simultaneously over $q$ times more dimensions $X^q$. This method of parallelization quickly becomes infeasible for even moderate dimensions and batch sizes. As before, adapting the method to conditional optimisation adds another layer of Monte-Carlo integration over $s \in S$ multiplying the computational cost.

Thompson sampling (TS) randomly suggests the next point to evaluate, $x^{n+1}$. TS also has the convenient mathematical property that $q$ step look ahead is equivalent to generating $q$ i.i.d samples [3, 4]. This property was used by [5] to parallelize MTS.

Alternatively, sequential construction of a batch can be done in $O(q)$ time and we consider the method of [6]. First $x^{n+1}$ is found by optimizing the chosen acquisition function $x^{n+1} = \arg\max \alpha(x)$. The acquisition function is then multiplied by a non-negative penalty function $\phi(x, x^{n+1})$ that penalizes $x$ similar to $x^{n+1}$. The next point is found by $x^{n+2} = \arg\max_x \alpha(x)\phi(x^{n+1}, x)$, then $x^{n+3} = \arg\max_x \alpha(x)\phi(x^{n+1}, x)\phi(x^{n+2}, x)$ until a batch of $q$ points is constructed. We use the inverted GP kernel as the penalty function

$$\phi((s, x), (s, x)^i) = 1 - \frac{k^0((s, x), (s, x)^i)}{k^0((s, x)^i, (s, x)^i)}.$$

See Figure 1 for an illustration. Previous work [7] showed that the conditional optimization setting is well suited to this construction method. Two points on dissimilar tasks do not interact and if they are both local peaks of the chosen acquisition function $\alpha(s, x)$, then both may be evaluated in parallel. In conditional optimization, the presence of task variables introduces multiple objective functions allowing a batch of points to be more spread out reducing interactions and possible inefficiencies. This can be achieved by penalization thus sidestepping the need for expensive nested Monte-Carlo integration. By contrast, in global optimization all $q$ points are "crammed" into a single task, all interacting with the same objective requiring more care in batch construction techniques.

This method for batch construction may be applied to any acquisition function, in our experiments we apply it to REVI and ConBO.

In practice, we optimize the acquisition function using multi start gradient ascent and keep the entire history of evaluations $\{(s_t, x_t, \alpha(s_t, x_t))\}_{t=1}^{\#calls}$. Since this history is very likely to contain multiple peaks, we simply apply the penalization to the set of past evaluations avoiding the need to re-optimize the penalized acquisition function. Therefore, efficiently parallelizing a conditional BO algorithm can be done in just a few additional lines of code.

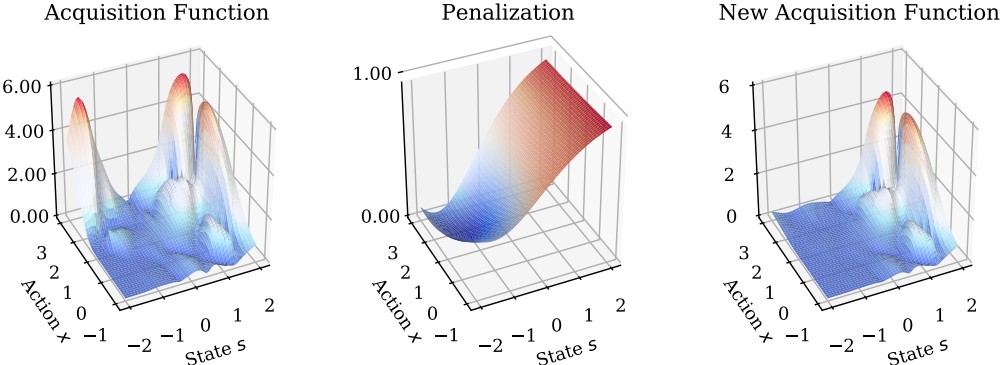

Figure 1: Left: acquisition function over $S \times X$ with a peak at $(s, x)^{n+1} = (-1.5, 2.5)$, the first point in the batch. Centre: the penalization function that down-weighs any point $(s, x) \in S \times X$ according to similarity with $(s, x)^{n+1}$. Right: The product of acquisition and penalization functions, the peak at $(s, x)^{n+2} = (1.6, 3.0)$ is the second point in the batch.

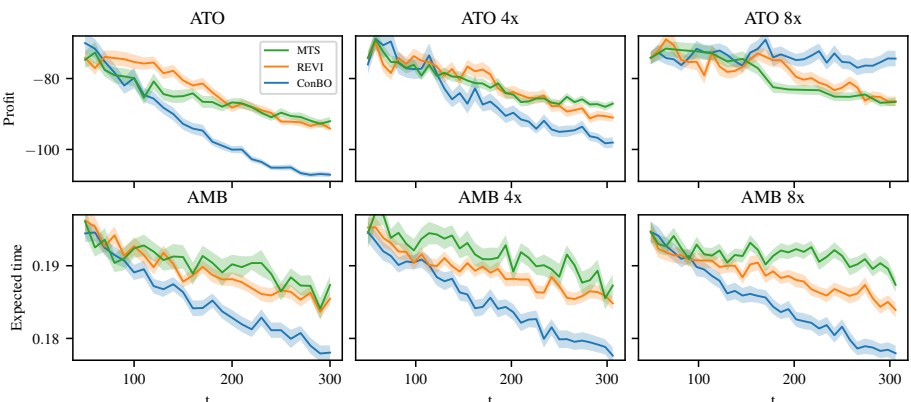

Figure 2: Algorithm performance on the Assemble To Order (top) and Ambulances (bottom) benchmarks. Serial, parallel 4 and 8.

# 5 Global Optimization

We perform a parameter sweep over the $n_z$ for each KG implementation. As baselines we considered Thompson sampling, Expected Improvement, Entropy Search and Random sampling and in these test problems Thompson sampling performed best and we use it as a baseline on all plots. For test functions we take the popular Branin-Hoo, Rosenbrock and Hartmann6. Thompson sampling is the best baseline and is compared to KG.

In future work we plan to perform rigorous comparison between a wider range of global optimization algorithms.

# 6 Contextual, Multi-Task and Multi-Fidelity Methods on Conditional Problems

We repeat experiments on the synthetic problems. Along with the unmodified ConBO and random search and apply the following methods to the synthetic benchmarks as shown in Figure 4.

- **ConBO-5 with random tasks** is the standard ConBO algorithm where the next task is sampled

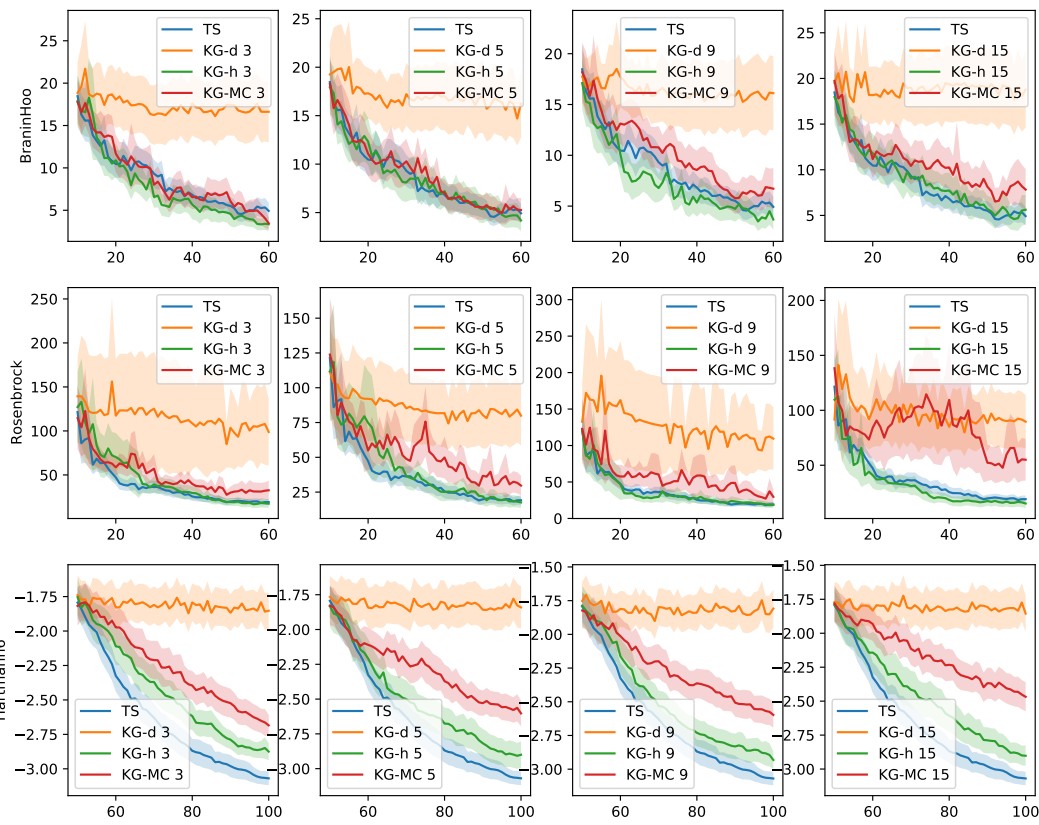

Figure 3: The KG by discretization, $KG_d$, uses $n_z$ random points in the search space $X$ and performs worst by far. $KG_{MC}$ uses $n_z$ samples of the future posterior mean function and performs well on small problems but suffers from unreliable noisy convergence. $KG_h$ consistently outperforms other KG methods and is the most reliable on all problems matching Thompson sampling but still performing worse on the Hartmann6 function. This is in contrast to the conditional setting with multiple tasks where integrated KG methods outperform Thomson sampling methods.

randomly $s^{n+1} \sim p[s] \propto W(s)$ and the next input is determined by maximising the acquisition function $x^{n+1} = \arg\max_x \text{ConBO}(s^{n+1}, x)$. This is simply a handicapped version of ConBO hence is converges significantly slower.

- **Bayesian Quadrature Optimization** [8] is an algorithm designed to find the input that is the best across the sum of all tasks weighted by $W(s)$. No modification is required to make such an algorithm apply to conditional optimization. As the data is collected for a different goal, the algorithm converges much slower than purpose made conditional optimization methods.

- **Multi-Information Source Optimization** [2] is an algorithm designed to optimize one target task $s^*$ given other cheap tasks. We set the target task $s^* = \arg\max W(s)$ and all tasks have equal cost. This algorithms greedily samples the single target tasks and all other tasks are ignored hence it never converges performing worse than random search.

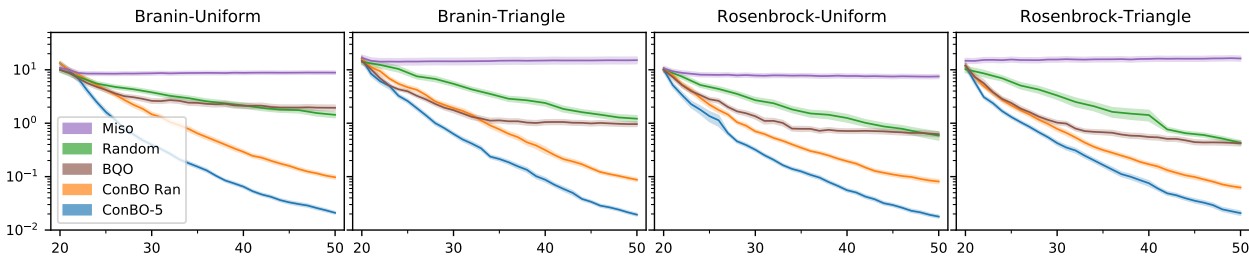

Figure 4: The algorithms not designed for conditional optimisation all yield significantly worse results.

# 7 Entropy Based Methods for Conditional Bayesian Optimization

Given a GP model and a dataset, $\mathbb{P}[x^*|\tilde{X}^n, Y^n]$ is the distribution over the peak of realizations of GP sample functions (abusing notation) $\mathbb{P}[x^*|\tilde{X}^n, Y^n] = \text{argmax}_x GP(\mu^n(x), k^n(x, x'))$. Given a new sample input $x^{n+1}$, the outcome $\mathbb{P}[y^{n+1}|x^{n+1}, \tilde{X}^n, Y^n]$ is also a random variable that is Gaussian. For thsi section, to reduce cluttering notation, we suppress the dependence on $\tilde{X}^n, Y^n$. The mutual information between random variables $y^{n+1}$ and $x^*$ is defined as

$$\text{MI}(x) = \int_{x^*} \int_{y^{n+1}} \log\left(\frac{\mathbb{P}[y^{n+1}, x^*]}{\mathbb{P}[y^{n+1}]\mathbb{P}[x^*]}\right) \mathbb{P}[y^{n+1}, x^*] dy^{n+1} dx^* \tag{39}$$

where $\mathbb{P}[y^{n+1}]$ depends upon $x^{n+1}$ yet we drop it for convenience.

## 7.1 Entropy Search

The Entropy search algorithm decomposes the above expression using $\mathbb{P}[y^{n+1}, x^*] = \mathbb{P}[y^{n+1}]\mathbb{P}[x^*|y^{n+1}]$ resulting in the following acquisition function

$$\text{ES}(x) = \int_{x^*} \log\left(\mathbb{P}[x^*]\right) \mathbb{P}[x^*] dx^* + \int_{y^{n+1}} \int_{x^*} \log\left(\mathbb{P}[x^* y^{n+1}]\right) \mathbb{P}[x^*|y^{n+1}] dx^* \mathbb{P}[y^{n+1}] dy^{n+1} \tag{40}$$

$$= H[x^*] - \int_{y^{n+1}} H[x^*|y^{n+1}] \mathbb{P}[y^{n+1}] dy^{n+1} \tag{41}$$

where $H[x^*]$ is the entropy of the distribution $\mathbb{P}[x^*]$. For the conditional case, the outcome $\mathbb{P}[y^{n+1}|(s, x)^{n+1}]$ is still a Gaussian random variable, and we measure the mutual information with the peak $x_{s_i}^*$ *over inputs*

*constrained to a given task* $\{s_i\} \times X$ that is $\mathbb{P}[x^*_{s_i}] = \operatorname{argmax}_x GP(\mu^n(s_i, x), k^n(s_i, x, s_i, x'))$. And the conditional entropy search acquisition function is simply

$$\text{ES}_c(s_i; (s, x)^{n+1}) = H[x^*_{s_i}] - \int_{y^{n+1}} H[x^*_{s_i}|y^{n+1}]\mathbb{P}[y^{n+1}]dy^{n+1}. \tag{42}$$

## 7.2 Predictive Entropy Search

We again drop the dependence on $x^{n+1}$ in $\mathbb{P}[y^{n+1}|x^{n+1}]$. The Predictive Entropy search algorithm uses an alternative decomposition of the Mutual Information using $\mathbb{P}[y^{n+1}, x^*] = \mathbb{P}[y^{n+1}|x^*]\mathbb{P}[x^*]$ resulting in the following acquisition function

$$\text{PES}(x) \quad = \quad H[y^{n+1}] - \int_{x^*} H[y^{n+1}|x^*]\mathbb{P}[x^*]dx^*. \tag{43}$$

For the conditional case, the outcome $\mathbb{P}[y^{n+1}|(s, x)^{n+1}]$ is still a Gaussian random variable, and we measure the mutual information with the peak $x^*$ *constrained to a given task* $s_i$ that is as above $\mathbb{P}[x^*_{s_i}] = \operatorname{argmax}_x GP(\mu^n(s_i, x), k^n(s_i, x, s_i, x'))$. And the conditional predictive entropy search acquisition function is simply

$$\text{PES}_c(s_i; (s, x)^{n+1}) = H[y^{n+1}] - \int_{x^*_{s_i}} H[y^{n+1}|x^*_{s_i}]\mathbb{P}[x^*_{s_i}]dx^* \tag{44}$$

where the expression $H[y^{n+1}|x^*_{s_i}]$ is the (non Gaussian) distribution of $y^{n+1}$ at $(s, x)^{n+1}$ given that the peak of task $s_i$ is at $x^*_{s_i}$.

## 7.3 Max Value Entropy Search

Max value entropy search instead measures the mutual information between the new outcome $\mathbb{P}[y^{n+1}|x^{n+1}]$ and the largest possible outcome (again abusing notation) $\mathbb{P}[y^*] = \max_x GP(\mu^n(x), k^n(x, x'))$, the peak value of posterior sample functions. The acquisition function decomposes the mutual information into

$$\text{MES}(x) = H[y^{n+1}] - \int_{y^*} H[y^{n+1}|y^*]dy^* \tag{45}$$

The conditional version measures the mutual information between $\mathbb{P}[y^{n+1}|(s, x)^{n+1}]$ and the largest $y$ value amongst all outcomes with the same task $\mathbb{P}[y^*_{s_i}] = \max_x GP(\mu^n(s_i, x), k^n(s_i, x, s_i, x'))$

$$\text{MES}(s_i, (s, x)^{n+1}) = H[y^{n+1}] - \int_{y^*_{s_i}} H[y^{n+1}|y^*_{s_i}]\mathbb{P}[y^*_{s_i}]dy^*_{s_i} \tag{46}$$