# OpenReview forum: "Practical Bayesian Optimization of Objectives with Conditioning Variables"
_NeurIPS.cc/2021/Conference — NeurIPS 2021 Submitted_

### Official Review · Reviewer_BDnT · 2021-07-15

**Rating:** 4
**Confidence:** 4

**Summary:**

The authors propose a method for conditional optimization problems: finding the optimum of each function in set of objective functions. In this setting, the method jointly selects the parameters to evaluate and the task to evaluate. The authors propose a theoretically-grounded variant of the Knowledge Gradient to address this class of problems and propose a new method of approximating the KG acquisition function.

**Limitations And Societal Impact:**

Limitations and societal impacts are discussed, but the potential negative societal impacts could be expounded upon.

**Main Review:**

Strengths:
-	Elegant, theoretically-grounded formulation for conditional optimization
  -	Being myopically optimal in the value of information framework is a nice property
-	The hybrid KG results show that is a quite stable method for approximating KG.

Weaknesses:
-	The class of problems is not very well motivated. The CIFAR example is contrived and built for demonstration purposes. It is not clear what application would warrant sequentially (or in batches) and jointly selecting tasks and parameters to simultaneously optimize multiple objective functions. Although one could achieve lower regret in terms of total task-function evaluations by selecting the specific task(s) to evaluate rather than evaluating all tasks simultaneously, the regret may not be better with respect to timesteps. For example, in the assemble-to-order, even if no parameters are evaluated for task function (warehouse s) at timestep t, that warehouse is going to use some (default) set of parameters at timestep t (assuming it is in operation---if this is all on a simulator then the importance of choosing `s` seems even less well motivated). There are contextual BO methods (e.g. Feng et al 2020) that address the case of simultaneously tuning parameters for multiple different contexts (tasks), where all tasks are evaluated at every timestep. Compelling motivating examples would help drive home the significance of this paper.
-	The authors take time to discuss how KG handles the continuous task setting, but there are no experiments with continuous tasks
-	It’s great that entropy methods for conditional optimization are derived in Section 7 in the appendix, but why are these not included in the experiments? How does the empirical performance of these methods compare to ConBO?
-	The empirical performance is not that strong. EI is extremely competitive and better in low-budget regimes on ambulance and ATO
-	The performance evaluation procedure is bizarre: “We measure convergence of each benchmark by sampling a set of test tasks S_test ∼ P[s] ∝ W(s) which are never used during optimization”. Why are the methods evaluated on test tasks not used during the optimization since all benchmark problems have discrete (and relatively small) sets of tasks? Why not evaluate performance on the expected objective (i.e. true, weighted) across tasks?
-	The asymptotic convergence result for Hybrid KG is not terribly compelling
-	It is really buried in the appendix that approximate gradients are used to optimize KG using Adam. I would feature this more prominently.
-	For the global optimization study on hybrid KG, it would be interesting to see performance compared to other recent kg work (e.g. one-shot KG, since that estimator formulation can be optimized with exact gradients)

Writing:
-	L120: this is a run-on sentence
-	Figure 2: left title “poster mean” -> “posterior mean”
-	Figure 4: mislabeled plots. The title says validation error, but many subplots appear to show validation accuracy. Also, “hyperaparameters” -> hyperparameters
-	L286: “best validation error (max y)” is contradictory
-	L293: “We apply this trick to all algorithms in this experiment”: what is “this experiment”?
-	The appendix is not using NeurIPS 2021 style files
-	I recommend giving the appendix a proofread:
  - Some things that jump out
    -	P6: “poster mean”, “peicewise-linear”
    -	P9: “sugggest”


**Time Spent Reviewing:**

2

---

> ### Author Response · Authors · 2021-08-10
> **Response to BDnT**
>
> Thank you for your review. We hope we can address each point.
>
> - For the cifar 10 experiment, our goal was to make a simple, cheap to run, easily reproducible benchmark, hence we use off-the-shelf data, off-the-shelf architecture, optimize a standard set of hyperparameters, and the experiment is easy to run, we used relatively weak Nvidia GTX 1070s from ebay. In contrast, a reinforcement learning application would require configuring both agents and environments, simpler "shallow" machine learning models, SVMs, logistic regression, random forests, are often cheap to run and complete hyperparamter tuning can be performed in minutes with basic methods negating the need for expensive black box methods. Bardenet et.al. 2013 used the AutoWeka dataset of datasets for hyperparamer sharing across datasets yet we struggled to find resources to reproduce the benchmark and deep learning has become much more popular in the meantime. The cifar10 experiment met all of benchmark design criteria and we hope it can be a useful tool for the community.
> - The warehouse problem has been considered in multiple previous Bayesian optimization works (cited) with a hard coded demand level $s_{constant}=1$, we generalize this to continuous demand levels $s\in [0.5, 1.5]$. Likewise the Ambulance problem has been considered with a hard coded population centre $s_{constant}=(20, 20)$ and we simply generalize this to continuous population centre $s\in [0, 30]^2$. The use of simulators is common practice in simulation optimization and Bayesian Optimization (we further cite applications in fluid dynamics, nuclear physics).
> - all benchmarks except for the cifar10 experiment, that is Branin Hoo, Rosenbrock, Warehouses, Ambulances, all use a continuous task setting, they are simply treated as black box functions with continuous arguments. While they can be discretized (this is how the REVI algorithm works), we consider the more general continuous case. It is for this reason we use separate test tasks to measure performance, this removes any chance of overfitting and ensures that the GP model generalizes across tasks.
> - in the finite task case with, say 10 tasks, sampling all tasks for a single $x$ consumes 10 samples in one go. Sequentially allocating 10 samples means that each sample gets the benefit of outcomes of all previous samples, this benefit is provable in a value of information framework, i.e. pre-allocation is guaranteed to be worse than sequential allocation (this assumes continuous outputs and Gaussian likelihood, this results does not apply to binary outputs and Bernoulli likelihood). As mentioned in response to other reviewers, many methods may be applied, eg. random search, they will not be as efficient and take much longer to converge, or reach inferior results in a given budget.
> - Regarding entropy methods, we felt that proposing Hybrid KG along with three algorithms, ConBO-KG, ConBO-ES, ConBO-MES, would be beyond the scope of the paper and we leave the final two methods to future work. We do not claim that ConBO-KG is the true best method and expect that ConBO-ES and ConBO-MES to perform conmprarbly depending upon application. The one main advantage of ConBO-KG is that KG acquisition function itself is a generalization of expected improvement and it is measured in units of function output, i.e. if $f(s, x)$ is in dollars, $KG(s_i; x,s)$ is also dollar improvement and ConBO-KG is sum of dollar improvements over tasks. Meanwhile ConBO-ES and ConBO-MES would be a sum of Shannon information improvements over tasks and is thus arguably less intuitive.
> - EI on the ATO and Ambuance benchmarks, we beleive, can be viewed as an outlier. The EI baseline does not use $W(s)$, it does not aim to optimize all tasks but only the task that contains the highest output. As there is no undeerlying theoretical justification for this result, we hypothesize that this is applications specific (i.e. EI "got lucky" on those benchmarks). We include it to highlight that such results do happen in practice and also that ConBO-KG was the only baseline that was not significantly worse than EI while the other conditional BO methods were.
> - we present the asymptotic convergence result as a peace-of-mind property of ConBO, we view this is a fundamental sanity check that demonstrates ConBO is not a heuristic . We can remove it if the reviewer feels so.
> - the gradients of ConBO-KG are analytically computed and as hybrid KG is an (extremely tight) lower bound to the true KG, the gradient vector cannot strictly be interpreted as an unbiased estimate of the gradient of the true KG, it is a gradient of a tight lower bound to the true KG hence "approximate". Note that there is a mistake in the SM, we state "approximate gradients can be computed assuming $X^*$ is fixed" however this assumption is not required and we have deleted this assumption and added the following proof
> $$
> KG_h(x) = KG_d(\mu(X^*(\underline{Z}, x)), \sigma(X^*(\underline{Z}, x), x))
> $$
>
> $$
> \frac{d}{dx}KG_h(x) = \underbrace{\frac{d}{dX^*}KG_d(\cdot)}_{=0} \frac{dX^*}{dx} + \frac{\partial}{\partial\sigma}KG_d(\cdot) \frac{\partial\sigma(X^*(\underline{Z}, x), x)}{\partial x}
> \\
> = \frac{\partial}{\partial\sigma}KG_d(\cdot) \frac{\partial\sigma(X^*(\underline{Z}, x), x)}{\partial x}
> $$
> where the gradient w.r.t. $X^*$ vanishes as $X^*$ is already optimal (this is proven too but left out of this response) and the final term is exactly the gradient used in the source code and all experiments and does not assume that $X^*$ is fixed. Hence any gradient approximation error is purely a result of using a lower bound.
> - all KG methods, Monte Carlo, discrete, one-shot, hybrid, approximate the same quantity, the true KG, and with infinite computation time will all perform identically. Methods only differ by their bias/variance for a given computational budget we explore this relationship in Section 3.1 where each method has the same computational budget. For Hybrid KG, this true KG value approximation error is much smaller than other methods, so will be the gradient though we have not explicitly shown gradient errors. Since all KG methods are _theoretically_ identical in the large computation limit, we do not perform extensive global optimization experiments in this work. We instead emphasize that the KG (or ES or MES) implementation inside ConBO must be as cheap and accurate as possible to avoid unreasonable overhead and hence ConBO is a critical use case for hybrid KG.

---

### Official Review · Reviewer_QbMn · 2021-07-16

**Rating:** 5
**Confidence:** 3

**Summary:**

The authors tackle an underexplored problem they call conditional BO, which is a variant of multi-task BO in which one seeks to identify the optimum of each task, instead of the traditional multi-task BO seeks to identify the optimum of the average over all tasks.

The authors introduce ConBO, which uses the knowledge gradient acquisition function to determine the next point to evaluate, and discuss its efficient computation through a combination of intelligent sampling and discretization.

ConBO is evaluated on a variety of problems, including three good applications: hyperparameter optimization, ambulance base placement, and assemble-to-order policy selection. Results are promising, and suggest that ConBO outperforms a well-chosen set of sensible baselines.

**Limitations And Societal Impact:**

Yes

**Main Review:**

Strengths:
* The applications were well suited to the problem at hand.
* The discussion of prior work and baseline performance measures was very comprehensive.
* The problem itself is both of theoretical interest and practical value.
* I also appreciated the author’s efforts to speed up the acquisition function computation; this is too often ignored in the literature as of minimal importance.

Weaknesses:
* The primary weakness, which the authors themselves point out, is that this problem of conditional BO is nearly identical to the problem of Bayesian quadrature; though the acquisition function itself is novel to my knowledge, the problem formulation is not.
* The acquisition function is the expectation of the KG acquisition over tasks s $E_s[ KG(s)]$, where the expectation is taken w.r.t measure W(s). My intuition says this acquisition seeks to efficiently compute $E_s[ x(s)]$ where $x(s) = \max_s \mu(x)$ w.r.t. W(s), which seems to be precisely Bayesian quadrature (please correct me if my intuition is wrong!)
* Thus, any method for Bayesian quadrature ought to work well, unless I am missing something. There don’t seem to be any Bayesian quadrature methods in the baselines, is there a reason why?
* I think this paper can be greatly strengthened if the authors define different measures of success. Obviously there is no way to identify exactly x(s) for all s, so the authors have decided their measure of success is $E_s[ x(s) ]$, which is one of the most logical. But there are other measures of success for conditional BO that do not necessarily involve this expectation; for example, computing accurately the bounds $\max_s | x(s)|$. Perhaps these can be used to showcase the strengths of ConBO, though this would likely require significant revision.


**Time Spent Reviewing:**

2

---

> ### Author Response · Authors · 2021-08-10
> **Response to QbMn**
>
> Thank you for your review.
>
> **Measure of Success**
>
> In global optimization (single task setting) one learns the location of the peak $x^*=\text{argmax}f(x)$ and the measure of success of this discovered location is the output value $f(x^*)$. In the setting we refer to as conditiona, there is a range of tasks, each one has a unique peak and we intend to find the location of the peak for each and every task
>
> $$
> x^*(s) = \text{argmax}_x f(s, x).
> $$
>
> Hence the measure of success of a single task is $f(s, x^*(s))$ and so the measure of success of all tasks is simply $\int_s f(s, x^*(s)) W(s) ds$, a natural generalization of the global optimization measure of success and is the quantity we report on all results plots. (As an aside, we believe the quantity $\mathbb{E_s}[x(s)]$ is not the goal of optimization, such a point may be viewed a single point in the input space $x_{centroid} \in X$ that is the "centroid" of all the locations of the peaks from all the tasks, we do not measure or report this result and such a result would not show how well each and every task has converged). Note that we consider noisy black box functions hence $f(s, x)$ is stochastic and the true measure of success is $\mathbb{E}_{\text{output noise}}[f(s, x^*(s))]$ and we try to state where expectations are over noise.
>
> Using $\max_s |x^*(s)|$ (or $\max_s|x^*(s) - x_{oracle}(s)|$) would return the biggest deviation between the learnt peaks and the origin (or the oracle ground truth peaks). We feel this is not a relevant measure of convergence of each and every task. Similarly, in global Bayesian optimization, it is rare to report $|x^* - x_{oracle}|$ as a convergence metric as it does not quantify the output value of the given point.
>
> **Multi Task BO and Bayesian Quadrature Optimization (BQO)**
>
> We do include Bayesian Quadrature Optimization  (multi task BO method) as a baseline in the Supplementary Material along with FABOLAS (multi fidelity method) and random task ConBO (heuristic contextual optimization method), we consider these to be baselines to be "off-topic" as they are designed for different problem settings. (EI is also an off topic baseline yet we include it as it is the most popular acquisition function and it occasionally "got lucky" and outperformed state of the art conditional baselines).
>
> Hence we feel this is simply a misunderstanding about the problem setting. For example, note that any global optimization method, such as EI, may be applied to a conditional problem. Such a method will collect data with the goal of finding global optimum over the (task, input) space, i.e. the single task with the single input that has the highest output
> $$
> x^*, s^* = \text{argmax} f(s, x)
> $$
> hence it will prioritize sampling the single task $s^*$ with and single input output $x^*(s^*)$ with highest output and neglect sampling other tasks $s\neq s^*$ and other tasks will take much longer to be optimized. Similarly, the goal of multi task (BQO) methods is to learn a single $x$ that is the input yielding the optimal compromise when applied to all tasks
> $$
> x^*= \text{argmax} \int_s f(s, x) W(s)ds
> $$
> Hence such methods prioritize collecting data around $x$ values for which many of the task $s$ have high output and will neglect sampling $x$ values that are optimal on one task $s_i$ and yet suboptimal on other tasks $s_j$, hence such task $s_j$ will again take a very long time to be optimized, $x^*(s_j)$ will be poor. As such, BQO will take a very long time to optimize each and every task. This is empirically demonstrated by the slower convergence in our results in the SM.
>
> We hope this clarifies any misunderstanding.

---

### Official Review · Reviewer_wodU · 2021-07-16

**Rating:** 5
**Confidence:** 4

**Summary:**

This paper proposes a new algorithm for a problem entitled "conditional Bayesian optimization." The goal is to search for a *series of input locations* that maximizes the performance of a given set of tasks. Unlike multi-fidelity optimization, the objective is to find good values for all tasks rather than the high-fidelity task. The main contribution is a Hybrid Knowledge Gradient (KG) algorithm that mixes two ways of computing KG: one that discretizes the input space (x, t) and computes KG for a discrete set of points,  and another one that uses the reparametrization trick to perform stochastic gradient descent over Monte Carlo samples. Formal guarantees about this method are also given. Empirical results show that the proposed method effectively reduces computation time and improves the quality of the solution.

**Ethical Concerns:**

N/A.

**Limitations And Societal Impact:**

Yes.

**Main Review:**

**Strengths**

I think this paper explores a problem formulation that is not well-study, but it could be relevant. It is a subtle variation of multi-task learning that focuses on multiple solutions (on for each task). This distinguishing is a little bit tricky, and I was a little bit confused for a while. Fortunately, the authors give three different examples that illustrate the setting that they are considering. I also enjoy the experiment section of this paper. The authors discussed the studied problems adequately, and I found the visualization and presentation of the results very interesting.

**Weaknesses**

Originality. This paper would be more substantial if the main algorithmic contribution weren't so tight to computational tricks to compute Knowledge Gradient or if the problem under investigation wasn't similar to REVI's [22] formulation.  If I understood correctly, the main contribution is a speedup over REVI based on how KG is calculated. The theoretical analysis adds substance to the paper, but none of these results are ground-breaking.

Clarity. The presentation of section 2 could be a little bit more straightforward. There is unnecessary confusion about how information is captured by the inference procedure (the Gaussian process) and how decision-making is handled (construct an acquisition function).

**Minors**

- 115 - "integrating over tasks multiplies the computational burden" multiples -> increases
- 130 - "non-trivial modification to be able to account for how a sample affects similar tasks." Wouldn't a kernel that captures the correlation between similar tasks be enough to accomplish this?
- 133 - location of the peak P(x | ...) -> mutual information between both distributions (or RV), you can remove the word peak here.
- 149 - I don't think you meant to say that squared exponential or Matern is a kernel that factorizes here.
- l89 - n_s? Shouldn't it be n_z?

**After rebuttal**
 I appreciate the author's detail response and I would strongly recommend the authors to add this level of clarity and detail to the original manuscript. Thanks

**Time Spent Reviewing:**

4

---

> ### Author Response · Authors · 2021-08-10
> **Response to reviewer wodU**
>
> Thank you for your review and concise summary.
>
> Regarding originality, the main novel contribution to the field of Bayesian optimization as a whole is a novel version of KG.
> However, in the literature that covers the same problem setting that we refer to as conditional optimization, many aspects have never been applied.
> - **significance of the proofs**: The PEQI method was designed for conditional optimization however we observe that it in fact does not converge. Likewise, Naively adopting a multi-fidelity method would also result in a method that doesn't converge. The proofs demonstrate that the proposed methods, Hybrid KG, ConBO, maintain their theoretical properties. Hence the proofs highlight that the modifications of known methods (KG, integration over tasks) would **not** result in an algorithm that fails to converge and also it does conform to the well known Value of Information methods, i.e. ConBO is not a heuristic.
> - **dynamic importance sampling of tasks**: it is a fundamental property of the problem setting that sampling one task will affect others. Previous methods either only account for this in the GP modelling part of BO and ignore it in the sampling part of BO (MTS, PEI. SCoT, PEQI) or REVI accounts for it through an (exponentially scaling) naive discretization approach. The current state of the art methods do not properly exploit the basic structure of the problem, i.e. "they are leaving performance on the table" which is particularly important for expensive black box optimization and should be exploited.
> - **stochastic gradient ascent**: as there is stochasticity in how the acquisition function is evaluated, we aim to minimize variance and bias in the estimates, hence we apply a range of methods (latin hypercube sampling, importance sampling, hybrid KG) to stabilize this as much as possible enabling application to much broader range of problems with much lower computational overhead. Again, state of the art methods do not consider such practicalities that can often cause an algorithm to perform poorly.
> - **experiments with a wide range of baselines**: in the current literature, we struggled to find (reproducible) open source benchmark problems. Therefore our secondary (unstated) goal in this manuscript is to provide practical advice and present a broad set of open source reproducible benchmarks highlighting both good and bad results. As a consequence, we observe that state of the art methods often are outperformed by even non-applicable methods (EI) while we ensure that ConBO does not suffer such failure modes.
>
> Minor comments:
>
>
> _"non-trivial modification to be able to account for how a sample affects similar tasks." Wouldn't a kernel that captures the correlation between similar tasks be enough to accomplish this?_
>
> BO methods require a GP model and an acquisition function. Previous works do fit a joint model over the (task, input) space (note that multi fidelity, multi task, contextual algorithms all apply similar modelling). However the acquisition function must also account for correlation across tasks (note that multi fidelity method **must** account for correlation also in acquisition). As stated above, this has only been briefly studied in the conditional literature with the REVI method that discretizes the task space and solves a discrete problem.
>
> Thank you so much for the rigorously reading the paper.

---

### Official Review · Reviewer_aWTB · 2021-07-17

**Rating:** 4
**Confidence:** 4

**Summary:**

This paper presents a framework for conditional optimization, which is called ConBO. Authors also proposed a new acquisition function: hybird knowledge gradient, which is built on KG. Empirical experiments show their proposed method outperform many competing algorithms.

**Ethical Concerns:**



**Limitations And Societal Impact:**

See the above review.

**Main Review:**

I reviewed this paper at AISTATS. While I was glad to see that authors have added additional comparison experiments on multi-task and multi-fidelity methods (but only in the supplementary material, not int the main paper), a lot of goods points raised by reviewers there are still not addressed in this version.

improvements:
1. many related work are covered in this version and this paper is well-organized

weaknesses:
1. Some related work is still missing.
2. There are insufficient updates in the main paper. Compared with the last version, all figures and empirical evaluation are exactly the same, without any changes. Although there are some extra experiments in the supplementary material.  Notably the comparison with [1] is still missing. This work is closely related to [1] and a comparison with [1] is recommended.
3. Given the proposed method doesn't provide sufficient improvements over KG, I would have expected stronger and extensive empirical evaluation.






[1]. Krause, A., Ong, C.S.: Contextual Gaussian Process Bandit Optimization. In: Shawe-Taylor, J., Zemel, R.S., Bartlett, P.L., Pereira, F., and Weinberger, K.Q. (eds.) Advances in Neural Information Processing Systems 24. pp. 2447–2455. Curran Associates, Inc. (2011).



**Time Spent Reviewing:**

2

---

> ### Author Response · Authors · 2021-08-09
> **The method of Krause et.al. was designed for a different problem**
>
> Thank you for reviewing the paper again.
>
> 1. Krause et.al. is cited, reference [25] in the paper.
>
> 2. Empirical evaluation Krause et.al.: we did not find an efficient way to adapt the method to the setting we consider. Equation [2] in Krause et.al. (adapted to our notation) give the acquisition function as follows
> $$
> x^{n+1} = \text{argmax}_x \quad \mu^{n}(x, s^{n+1}) + \sqrt{\beta^{n}}  \sigma(x, s^{n+1})
> $$
> and does not immediately yield a method to determine $s^{n+1}$, in the contextual setting, it is given at each iteration. The few obvious options are as follows
>
> - optimize for $s^{n+1}$ with $x^{n+1}, s^{n+1} = \text{argmax}_{x, s} \mu^{n}(x, s) + \sqrt{\beta^{t}}  \sigma(x, s)$
> however this method ignores $W(s)$ and hence will perform poorly in settings where $W(s)$ is non-uniform. Further, this method does not integrate over tasks. The same theoretical approach is captured in the baselines MTS and PEQI, both have a similar design motivation "find the one task that would benefit itself the most from a sample" and not "find the one task that can benefit everyone from a sample".
>
> - optimize for $s^{n+1}$ with $x^{n+1}, s^{t+1} = \text{argmax}_{x, s} W(s) \left( \mu^{n}(x, s) + \sqrt{\beta^{t}}  \sigma(x, s)\right)$ would imply that small W(s) has acquisition value towards zero, even if $\mu^{n}(x, s) + \sqrt{\beta^{t}}  \sigma(x, s) <0$ for all $x, s$. For example, for objective functions that lie below the y = 0 plane will sample locations where W(s) is small, precisely the opposite of the desired behaviour. This does not apply to PEQI or MTS (which have strictly non-negative acquisition functions) hence we did add a $W(s)$ weighting.
>
> - randomly sample $s^{n+1}\sim P[s]\propto W(s)$. Algorithmically, this is equivalent to using random search to determine task $s^{n+1}$ followed by using Bayesian Optimization to determine $x^{n+1}$. Hence we added such a method as a baseline in the supplementary material, however using ConBO as the $x^{n+1}$ acquisition function hence forming an ablation study with the standard ConBO.
>
> Hence, we believe that there are naive ways to "hotfix" the referenced algorithm to work, yet they do not significantly differ from current baselines. We acknowledge that such a hotfixxed method may work well in certain applications. We show that EI on the ambulance and warehouse problem can work well, however as there is no theoretical justification for such a result, we hypothesize that this is application specific and investigating such edge cases is beyond the scope of this work.
>
> 3. all methods for computing KG: discrete, Monte Carlo, hybrid KG, are approximations of the true theoretical KG value. Given infinite computational time they will all converge to the exact same true KG value, and hence they become the exact same algorithm and will perform exactly the same on any benchmark problem. These KG implementations only differ by their approximation accuracy fora  given computational cost. Hence in Section 3.1 and Table 1, we try to quantify the bias and variance of approximating the true KG for each  method for different computational costs, we see that hybrid KG can have significantly smaller bias, much lower than the variance or far more expensive methods. This bias can be reduced to negligible with very little computational cost.

---

### Decision · Program_Chairs · 2021-09-27

**Decision:**

Reject

**Comment:**

This paper addresses a framework for conditional Bayesian optimization with a newly proposed hybrid KG. The method seems to be a variant of multi-task BO, where the optimum of each task is determined in contrast to the multi-task BO. The applications well suited to the problem are described to demonstrate the usefulness of the method. Reviewers agree that the paper is of theoretically interest and has practical value as well. However, a few issues were raised, details of which can be found in the reviewers' comments. During the discussion period, none of the reviewers were willing to champion this work. The paper still needs more substantial updates in particularly experiments.